# Khayyam Challenge (PersianMMLU): Is Your LLM Truly Wise to The Persian Language?

**Omid Ghahroodi**♠◇, **Marzia Nouri**♠◇\*, **Mohammad Vali Sanian**♠◇\*, **Alireza Sahebi**♠◇\*,
**Doratossadat Dastgheib**♠◇, **Ehsaneddin Asgari**‡†, **Mahdieh Soleymani Baghshah**♠◇†,
**Mohammad Hossein Rohban**♠◇†

♠ Raia Center for Artificial Intelligence Research
◇ Computer Engineering Department, Sharif University of Technology, Iran
‡ Qatar Computing Research Institute, Qatar

{oghahroodi98, nouri.marzia.1999, mvs2667, arsahebi97}@gmail.com
 d_dastgheib@sbu.ac.ir, easgari@hbku.edu.qa, {soleymani, rohban}@sharif.edu

## Abstract

Evaluating Large Language Models (**LLMs**) is challenging due to their generative nature, necessitating precise evaluation methodologies. Additionally, non-English LLM evaluation lags behind English, resulting in the absence or weakness of LLMs for many languages. In response to this necessity, we introduce **Khayyam Challenge** (also known as **PersianMMLU**), a meticulously curated collection comprising **20,805 four-choice questions** sourced from **38 diverse tasks** extracted from Persian examinations, spanning a wide spectrum of subjects, complexities, and ages. The primary objective of the **Khayyam Challenge** is to facilitate the rigorous evaluation of LLMs that support the Persian language. Distinctive features of the **Khayyam Challenge** are **(i)** its comprehensive coverage of various topics, including literary comprehension, mathematics, sciences, logic, intelligence testing, etc aimed at assessing different facets of LLMs such as language comprehension, reasoning, and information retrieval across various educational stages, from lower primary school to upper secondary school **(ii)** its inclusion of rich metadata such as human response rates, difficulty levels, and descriptive answers **(iii)** its utilization of new data to avoid data contamination issues prevalent in existing frameworks **(iv)** its use of original, non-translated data tailored for Persian speakers, ensuring the framework is free from translation challenges and errors while encompassing cultural nuances **(v)** its inherent scalability for future data updates and evaluations without requiring special human effort. Previous works lacked an evaluation framework that combined all of these features into a single comprehensive benchmark. Furthermore, we evaluate a wide range of existing LLMs that support the Persian language, with statistical analyses and interpretations of their outputs. We believe that the **Khayyam Challenge** will improve advancements in LLMs for the Persian language by highlighting the existing limitations of current models, while also enhancing the precision and depth of evaluations on LLMs, even within the English language context.

---

\*These authors contributed equally to this work and are considered joint second authors. The order is listed randomly to reflect their equal contributions.

†These authors contributed equally to this work and are considered joint corresponding authors. The order of corresponding authors is listed randomly to reflect their equal contributions.

# 1 Introduction

**Large Language Models (LLMs)** have recently revolutionized applications of machine intelligence (Hong et al., 2024; Wu et al., 2023; Thirunavukarasu et al., 2023; Glukhov et al., 2023). The rapid deployment of these models within industrial and public sector solutions has made evaluating their capabilities an imperative task (Guo et al., 2023). To address critical aspects of language understanding and bridge the gap between the knowledge that models observed during pretraining and the measures of success, the MMLU benchmark was introduced (Hendrycks et al., 2021). It assesses the ability of LLMs across a diverse set containing 57 subjects.

Although automatic translation efforts like MMLU can bridge the gap in evaluation resources for languages other than English (Achiam et al., 2023), the cultural specificity of these questions limits their applicability across different languages and cultures. This is particularly true for Persian, a language with its own rich culture and literature, where direct translation of English MMLU content may not be suitable for certain culture-specific subjects. In addition, our investigation reveals significant complications in automatic translation, particularly in specialized domains like Physics, which can be attributed to the quality of state-of-the-art translations for specific subjects. Despite advancements in machine translation technology, accurately conveying precise meaning in certain domains remains a hard task. For instance, the GPT-3.5 automatic translation often fails to include essential units such as "microjoules" in translations, leading to inaccuracies. Also, loss of information can occur in translation due to the existence of Persian words that lack exact equivalents in English. For example, in Persian, there are distinct terms for paternal aunt, maternal aunt, paternal uncle, and maternal uncle, while in English, only "aunt" and "uncle" are used (See Table 4). This underscores the inherent difficulty in translating domain-specific terminology accurately. For such reasons, some studies have evaluated LLMs on original non-English datasets rather than the translated ones. For instance, Li et al. (2023) has introduced a Chinese dataset across 67 topics and indicated that current models struggle to achieve accuracies above a certain threshold. Also, Zhang et al. (2023) introduced a multilingual and multimodal dataset, and showed that multilingual text processing hardly achieves over 60% accuracy.

Motivated by the mentioned issues, we propose the Khayyam Challenge, also referred to as PersianMMLU, a benchmark designed to analyze the performance of LLMs in Persian and evaluate their knowledge and abilities comprehensively. Named in honor of the famed Persian polymath Omar Khayyam, whose contributions spanned various disciplines including Mathematics, Astronomy, Philosophy, and Poetry, the Khayyam Challenge embodies the multidimensional nature of Persian language understanding. This benchmark covers 38 subjects, including Mathematics and Physics, which require reasoning and computational ability, to Humanities and Social Sciences, demanding nuanced understanding and cultural sensitivity. Unlike the previous Persian datasets such as ParsiNLU (Khashabi et al., 2021), our benchmark includes more diverse topics in addition to different educational stages. Moreover, our dataset distinguishes itself from previous efforts like ParSQuAD (Abadani et al., 2021) by being originally constructed in Persian, naturally incorporating the nuanced semantics and intricacies inherent to the language itself, rather than solely focusing on literal translations. Unlike PersianQA (Ayoubi, 2021) and PQuAD (Darvishi et al., 2023), which are extractive datasets where models are tasked with extracting answers from given paragraphs and questions, our benchmark offers a more comprehensive evaluation of LLMs. This is because the task of answer extraction alone may not sufficiently assess the models' overall language understanding and reasoning capabilities. Our proposed dataset contains "Iran's national university entrance exam", and Kanoon Farhangi Amoozesh (Cultural Educational Center), wherein questions are accompanied by metadata for each question. This metadata includes the difficulty level, a descriptive answer, the educational stage, the subject, and the specific topic of the question. Through the Khayyam Challenge, we aim to provide a holistic evaluation framework that reflects the diverse linguistic and cognitive challenges inherent in processing Persian text across various domains.

In our evaluations, we assessed several state-of-the-art language models, including GPT-3.5, GPT-4 (OpenAI, 2023), Aya (Üstün et al., 2024), PersianMind (Rostami et al., 2024), mGPT

(Shliazhko et al., 2022), mT0 (Muennighoff et al., 2022), Claude3-haiku (Anthropic, 2024), and XVERSE[1], all purportedly equipped with some level of understanding of the Persian language. Our findings indicate that while most of these models struggle to grasp Persian nuances, particularly evident in domains such as Calculus, Logic, and Geometry where accurate comprehension is essential, some exhibit comparatively better performance in contexts reliant on contextual understanding, such as Economics, Psychology, and Social studies. Notably, GPT-4 showcased relatively improved performance across multiple domains. However, there remains a clear imperative need for further enhancements across all models, especially in technical disciplines like Discrete Mathematics, where precise language comprehension is paramount for meaningful outcomes.

Furthermore, our evaluation unveiled new insights. We observed dependencies between the perceived difficulty of questions for humans versus LLMs. While both encounter similar challenges in tackling difficult questions, LLMs like GPT-4 demonstrated superior accuracy compared to humans, in questions intentionally designed to deceive, referred to as trapped questions. Moreover, our analysis identified biases in certain language models' responses, such as GPT-3.5 favoring particular choices, suggesting room for further improvement in these models.

Overall, the Khayyam Challenge (PersianMMLU) marks a significant step forward in evaluating the language understanding and abilities of LLMs that support the Persian language.

Our dataset and code are available on HuggingFace[2] and GitHub[3], respectively. Additionally, we set up a leaderboard[4] on HuggingFace to stay updated with the performance of other models.

## 2 Related work

### 2.1 Large Language Models

Over the past few years, there has been a significant improvement in the performance of language models. This progress has been observed in line with the scaling law (Kaplan et al., 2020), thanks to the increasing size of training datasets, enhanced processing power, and new evolved model architectures. The continuing process of scaling the models resulted in LLMs like GPT-3 (Brown et al., 2020), GPT-4 (OpenAI, 2023), Claude3, mT0 (Muennighoff et al., 2022), XVERSE, Aya (Üstün et al., 2024), etc.

Even though AI models are highly capable of solving various tasks, they continue to encounter difficulties when it comes to real-world problems that, for example, require strong reasoning abilities or complex mathematical calculations (Chang et al., 2024; Zhong et al., 2023). Therefore, we need to assess the effectiveness of these models in solving high-level tasks. This enables us to identify the weak points of the models and work towards improving them in the future.

Despite some of the recent LLMs being multilingual, studies indicate that their effectiveness is not as pronounced in non-Latin or low-resource languages as it is in English (Zhang et al., 2023). Consequently, it is essential to assess multilingual LLMs on tasks that employ languages other than English.

### 2.2 Evaluation of LLMs

Several benchmarks have been developed to assess the performance of LLMs. One of the most significant benchmarks is MMLU (Hendrycks et al., 2021), which evaluates language models for answering multiple-choice questions in 57 different tasks, but only in English.

---

[1]https://github.com/xverse-ai
[2]https://huggingface.co/datasets/raia-center/khayyam-challenge
[3]https://github.com/raia-center/khayyam-challenge
[4]https://huggingface.co/spaces/raia-center/PersianMMLU

| Benchmark | Languages | Type | NLU Tasks | # Instance | Metadata Desc. Ans. | Diff. Lev. | Trap | # Task |
|---|---|---|---|---|---|---|---|---|
| MMLU | Eng. | orig. | MCQA | 15,908 | ✗ | ✗ | ✗ | 57 |
| AGIEval | Eng., Chi. | orig.+trans. | MCQA, FIB | 8,062 | some instances | some instances | ✗ | 20 |
| M3Exam | 9 (no Persian) | orig. | MCQA | 12,317 | ✗ | = edu. stages | ✗ | 4 |
| ParSQUAD | Persian | trans. | RC | 70,560 | - | ✗ | - | 477 titles |
| PersianQA | Persian | orig. | RC | 9,938 | - | ✗ | - | 991 titles |
| PQuAD | Persian | orig | RC | 80,000 | - | ✗ | - | 19 |
| ParsiNLU | Persian | orig.+trans. | MCQA, RC, SA, TE, QP, MT | 14,500 | ✗ | ✗ | ✗ | 3 in MCQA |
| Khayyam Challenge | Persian | orig. | MCQA | 20,805 | ✓ | ✓ | ✓ | 38 |

Table 1: Comparison of various features of Persian and English benchmarks. As descriptive answers and trapped questions were not defined for reading comprehension benchmarks, we marked those fields with a hyphen (-). Desc. Ans.: Descriptive Answer, Diff. Lev.: Difficulty Level, Eng: English, Chi: Chinese, orig.: original non-translated question, tran.: translated question, MCQA: Multiple-Choice Question Answering, FB: Fill in the Blank, RC: Reading Comprehension, SA: Sentiment Analysis, TE: Textual Entailment, QP: Question Paraphrasing, MT: Machine Translation., edu. stages: educational stages

M3Exam (Zhang et al., 2023) introduces a multilingual, multimodal, and multilevel benchmark for evaluating LLMs including more than 12K multiple-choice questions from 9 languages (excluding Persian) at three educational stages. AGIEval (Zhong et al., 2023) is another benchmark that assesses the performance of LLMs on human-centric standardized exams in English and Chinese languages to measure their ability in human-level tasks.

There have been a few benchmarks built to assess language models on the Persian language, including ParSQUAD (Abadani et al., 2021), PersianQA (Ayoubi, 2021), ParsiNLU (Khashabi et al., 2021), and PQuAD (Darvishi et al., 2023), in which some of their features are compared in Table 1. ParSQUAD, PersianQA, and PQuAD present extractive datasets where models are asked to extract answers from given paragraphs. While this task can gauge models' reading comprehension skills, it may not effectively evaluate general capabilities and inherent knowledge of models. ParsiNLU evaluates language models based on 14500 questions from six language understanding tasks, including multiple-choice QA (MCQA), sentiment analysis, and more (Table 1). However, the questions in this benchmark fail to reach human-level complexity, thus inadequately assessing important skills of LLMs, including complex reasoning, needed for solving higher educational stages questions. Moreover, ParsiNLU covers only three subject tasks in MCQA and lacks adequate metadata, such as question difficulty levels. This deficiency further restricts our capacity to evaluate the model's proficiency across specific tasks and different difficulty levels.

To address this gap, we introduce Khayyam Challenge, which features multiple-choice questions sourced from high-standard exams, and contains reach metadata to evaluate the innate knowledge and human-like skills of LLMs across different difficulty levels and educational stages.

## 3 Data

The Khayyam Challenge presents a robust dataset aimed at enhancing the evaluation of LLMs that support Persian, particularly in the context of multiple-choice question answering. This dataset encompasses a diverse range of subjects, reflecting a comprehensive approach to assessing various cognitive abilities including language comprehension, reasoning, and knowledge recall across different educational stages.

The educational system in Iran, from which this dataset draws, is structured into 12 years of schooling divided into segments: 6 years of primary education and 6 years of secondary education. Primary education is split into lower primary school (LPS) for the first 3 years, followed by upper primary school (UPS) for the next 3 years. Secondary education is similarly divided, with lower secondary school (LSS) encompassing the first 3 years, and upper secondary school (USS) comprising the final 3 years.

### 3.1 Data construction

The dataset is sourced from the "Pellekan Yadgiri (Learning Ladder)" website[5], a part of the Kanoon Farhangi Amoozesh[6] (Cultural Educational Center), an educational institution in Iran. The platform provides resources for creating and administering tests, as well as standardized solutions for exercises. The dataset includes questions from various educational stages and subjects, including items from the Iranian national university entrance examination. This diversity contributes to the dataset's comprehensiveness and relevance.

While the dataset is primarily sourced from the Learning Ladder platform, we implemented additional processing steps. These included parsing HTML to extract attributes, filtering out questions with images or tables, removing explanatory answer questions, and deduplicating entries. We then reviewed the questions, corrected typos, and deleted those that required context that was not present in the question itself. These steps were crucial in refining the raw data into a structured, text-only dataset suitable for our research purposes.

We also note that the Learning Ladder platform serves as a platform for questions contributed by a diverse group of educators from various backgrounds and disciplines in education. This diversity helps mitigate individual biases. Furthermore, the inclusion of questions from the university entrance exams, carefully selected from submissions by top educators, adds an additional layer of quality control. These features minimize the presence of potential biases in PersianMMLU. This dataset is distributed under a Creative Commons No Derivatives (CC ND) license, prohibiting the creation of derivative works.

We also tested the hypothesis that the data from this collection was included in the training data of the models based on Oren et al. (2024). We conducted this evaluation using the Aya model, the best publicly available model, which had access to the token probabilities. The hypothesis that the evaluation data was included in the training data was not proven, with a p-value of 0.3909.

### 3.2 Metadata characteristics

The Khayyam Challenge is enriched with valuable metadata that elevates its utility beyond a simple aggregation of questions. This metadata includes:

- **Educational Stage**: Specifies the educational stage for which the question is intended (LPS, UPS, LSS, USS), allowing for the assessment of appropriateness and difficulty relative to the expected knowledge base at each stage of education.

- **Difficulty Level**: Each question is classified into one of five distinct difficulty levels: easy, relatively easy, medium, relatively difficult, and difficult. This nuanced categorization allows a detailed analysis of question difficulty and examinee performance. The difficulty level, like all metadata attributes, was extracted directly from the Learning Ladder platform. The platform uses an automated system to classify question difficulty based on student performance data. Questions answered incorrectly by a higher percentage of students are marked as more difficult.

- **Descriptive Answers**: In addition to the correct answer, our dataset provides a detailed explanation for each question. This is crucial for understanding the reasoning behind the correct answer, facilitating a deeper comprehension of the question.

- **Trap**: Some questions contain a "trap" choice—an incorrect answer that might be easily mistaken for the correct one. These questions are referred to as "trapped questions" and are generally more challenging, with the majority classified as difficult. This helps in understanding common misconceptions and the effectiveness of question design in truly testing knowledge and reasoning abilities. This attribute is also determined by the Learning Ladder platform, which analyzes frequently selected incorrect options to identify common misconceptions among students.

---

[5]https://peleyad.com/
[6]https://en.kanoon.ir/

- **Candidate Choice Distribution**: This metric provides the percentage of students selecting each answer choice for a given question.

- **Specific Topic**: Questions are meticulously categorized into detailed subjects, such as "Mathematics > Discrete Maths > Combinatorics." This detailed classification enables targeted analysis of exam content and provides insights into the distribution and depth of questions across various subjects.

- **Year**: Indicates the year when the question was designed, which can provide insights into the evolution of question complexity and educational standards over time.

The inclusion of this metadata is not merely for augmentative purposes; it serves a critical role in enabling comprehensive analyses that can benefit educators, researchers, and developers of educational technologies. Specifically, it allows for the comparison of performance between human examinees and LLMs on specific topics under varying difficulty levels. By assessing whether LLMs fall for the traps as humans or how they approach questions requiring complex thought processes, we can gain valuable insights into the capabilities and limitations of current AI technologies in educational contexts.

Moreover, the presence of descriptive answers supports the development of more sophisticated AI models by facilitating "chain of thought" processing, where the model learns to approach a problem step-by-step, mirroring human problem-solving methods. This not only enhances the model's problem-solving skills but also its ability to explain its reasoning in a manner that is understandable to humans.

### 3.3 Data statistics

The dataset contains 20,805 multiple-choice questions across 38 tasks, spanning subject areas like humanities, mathematics, natural science, and social science, along with elements of intelligence testing. These questions necessitate a blend of knowledge and reasoning. Additionally, the dataset includes 16,955 questions with human performance data, excluding Iran's national university entrance exam questions, and features 3,889 trapped questions. Figure 3-(a) depicts the allocation of questions among the main categories and their respective tasks. Figure 3-(b) outlines the distribution of questions based on their levels of difficulty. For more detailed information about the data, refer to Appendix A.

### 3.4 Key features

The Khayyam Challenge Dataset stands out for several reasons:

- **Comprehensive Coverage**: It spans a broad spectrum of subjects from literary comprehension to logic and intelligence testing, catering to different stages of education. This diversity makes it a versatile tool for assessing language models' capabilities across various domains.

- **Rich Metadata**: The inclusion of detailed question metadata enhances the dataset's utility for nuanced analysis and model evaluation, providing valuable context for each question.

- **New Data Utilization**: By incorporating questions never before used in research, the dataset avoids common data contamination issues, offering a fresh challenge to language models.

- **Original, Non-Translated Content**: Focused on the Persian language, the dataset eliminates translation errors and incorporates cultural nuances, making it uniquely valuable for related linguistic and cultural studies.

- **Scalability**: The dataset's design and sourcing methodology ensure its adaptability and expandability, allowing for straightforward updates and extensions without substantial human intervention.

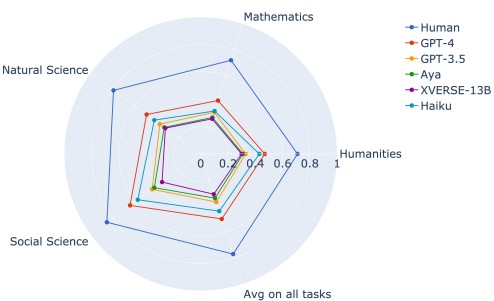

Figure 1: Comparison of accuracy across main categories for humans and various models

## 4 Experiments

### 4.1 Settings

We evaluated our benchmark using nine distinct models: GPT-4 (gpt-4-0613), GPT-3.5 (gpt-3.5-turbo-0125), Aya, XVERSE-13B, XVERSE-7B, PersianMind, mT0XL, mGPT, and Claude 3/Haiku. Detailed descriptions of each model can be found in Appendix B. Additionally, we benchmarked a standardized prompt-0 template (Figure 5) on the entire dataset and two other templates, prompt-1 (Figure 6) and prompt-2 (Figure 7), on a subset of 1000 samples. We also conducted Chain-of-Thought (CoT) (Figure 8) on a subset of 1000 samples requiring CoT, such as mathematical questions. To ensure consistency and fairness throughout our experiments, we kept all model hyperparameters at their default values. Furthermore, we set the temperature parameter to zero for all models and did not impose any maximum limit on the number of tokens, allowing models the freedom to conduct any type of inference they desired.

### 4.2 Answer extraction methods

The input prompt did not restrict the model to single-letter answers, as it might hinder the reasoning process. Therefore, we employed three distinct methods to extract the desired answer choice from the model output; "Regex", "Single Token Probability", and "Full Answer Probability".

In the "Regex" method, we developed detailed regex patterns tailored to each model to accurately capture the desired choice from the model output (the patterns are available in our GitHub repository). In cases where regex patterns failed to identify the desired choice, we utilized a pre-trained language model to generate embedding vectors for each of the answer choices and the model output. Finally, the choice with the most similar vector to the output's vector is selected as the desired answer.

In the "Single Token Probability" method, applied to models providing token probabilities, we used the probability distribution for the initial token in the model output to extract the probabilities associated with answer choice labels (e.g., 'a', 'b', 'c', 'd' in prompt-1 template (Figure 6)). The label with the highest probability is selected, leading to the choice of its corresponding option as the desired answer.

In the "Full Answer Probability" method, we calculated the average of the logarithm of token probabilities for each answer choice by using the probability distribution of the initial token of the model output. We then chose the answer with the highest average as the desired option. This process is shown in equation 1.

$$\text{Answer} = \arg\max_i \left( \frac{1}{n_i} \sum_{j=1}^{n_i} \log(p_{ij}) \right) \qquad (1)$$

Our analysis shows that the "Regex" method has the highest accuracy, whereas the "Full Answer Probability" method demonstrated a significant decrease in accuracy compared to the other two approaches.

### 4.3 Measure human performance

To measure human performance, we used the percentages of respondents who selected each answer choice for every question, as provided in the metadata. A question is labeled as correctly answered if the percentage selecting the correct answer exceeds the combined percentages of the other options. Otherwise, it is labeled as incorrectly answered. Human accuracy is then evaluated by counting the number of correctly and incorrectly answered questions.

### 4.4 Impact of translation quality

We aim to assess the impact of translation from state-of-the-art (SOTA) translation models on performance. To do so, we selected a set of 50 sample questions from three subjects of our dataset and translated them to English with the assistance of field experts, as well as using off-the-shelf translation models. Following this, we evaluated the GPT-4's performance on both sets of samples: those translated with expert assistance and those translated by models alone. Our findings which are illustrated in Table 2a, revealed a notable decrease in performance when translated with the help of off-the-shelf models, highlighting the need for a new dataset that does not depend solely on translated data. We also evaluated the impact of automatic translation on performance by translating a sample of English questions from the MMLU dataset (Hendrycks et al., 2021) into Persian and measuring GPT-3.5's performance on both sets. As shown in Table 2b, automatic translation from English to Persian can reduce the performance of large language models.

| PersianMMLU Topics | En acc. | En* acc. | Fa acc. |
|:---:|:---:|:---:|:---:|
| Physics | 58 | 76 | 50 |
| Chemistry | 74 | 82 | 86 |
| Math | 50 | 52 | 56 |

(a)

| MMLU Topics | En acc. | Fa acc. |
|:---:|:---:|:---:|
| Physics | 42 | 26 |
| Biology | 76 | 74 |
| Machine Learning | 46 | 40 |

(b)

Table 2: Accuracy rates of LLMs for **(a)**: a sample of questions from our dataset (Persian-MMLU) and their translations to English using an automatic translation model, and a field expert (denoted with an asterisk), and **(b)**: a sample of MMLU English questions and their translation to Persian using automatic translation model

### 4.5 Limitations of few-shot approach

We have developed a benchmark code for our dataset and used the zero-shot and CoT methods to calculate accuracy. Previous studies (Li et al., 2023; Zhong et al., 2023; Zhang et al., 2023) shows that using few shots on instruction-tuned models does not enhance accuracy and may even may decrease it. Therefore, we did not measure accuracy using few-shot techniques.

| Main Categories | Human | GPT-4 | Haiku | GPT-3.5 | Aya | XVERSE-13B | XVERSE-7B | PersianMind | mT0XL | mGPT | Random |
|:---:|:---:|:---:|:---:|:---:|:---:|:---:|:---:|:---:|:---:|:---:|:---:|
| Humanities | 0.71 | 0.47 | 0.43 | 0.33 | 0.31 | 0.3 | 0.27 | 0.28 | 0.28 | 0.25 | 0.25 |
| Mathematics | 0.72 | 0.41 | 0.33 | 0.32 | 0.28 | 0.27 | 0.28 | 0.26 | 0.26 | 0.25 | 0.25 |
| Natural Science | 0.79 | 0.49 | 0.42 | 0.37 | 0.33 | 0.32 | 0.29 | 0.29 | 0.28 | 0.25 | 0.25 |
| Social Science | 0.85 | 0.64 | 0.57 | 0.44 | 0.42 | 0.35 | 0.33 | 0.32 | 0.33 | 0.24 | 0.25 |
| Avg on all tasks | 0.77 | 0.5 | 0.44 | 0.37 | 0.34 | 0.31 | 0.29 | 0.29 | 0.29 | 0.25 | 0.25 |
| Avg on all questions | 0.76 | 0.49 | 0.41 | 0.36 | 0.32 | 0.3 | 0.29 | 0.28 | 0.28 | 0.25 | 0.25 |

Table 3: Human vs model accuracy for main categories: model answers extracted via "Regex"

# 5   Results and discussions

## 5.1   Results across all models

The evaluation results of the zero-shot method for all models across five main categories and three choice-extraction methods ("Regex", "Single Token Probability", and "Full Answer Probability"), as well as human performance, are presented in Table 3, and appendix Tables 9, 8. The more comprehensive results on all 38 tasks are reported in the appendix (Tables 10, 11, 12). Also, the results of CoT and its comparison with zero-shot for GPT-3.5 using "Regex" method across three main categories on a subset of dataset with 1000 questions are presented in appendix Table 7. These results yield the following key findings:

- Utilizing "Regex" method for answer extraction (Table 3) results in highest model performance compared to "Single Token Probability" (Table 9) and "Full Answer Probability" (Table 8) due to its more accurate and comprehensive choice extraction procedure. In the rest of the results section, we compare the accuracy of models using the "Regex" method.
- GPT-4 outperforms all other models in all five main categories, with an average accuracy of 6 percent higher than Claude3-haiku, the second-best performing model.
- Aya, an open-source model, performs comparably or even better than GPT-3.5, a closed-source model, in 4 tasks including Sociology USS and Economy USS. This demonstrates the convergence of open-source models' capabilities towards closed-source models.
- Although PersianMind, a 7B Persian-English LLM, is trained and fine-tuned on 2 billion Persian tokens (Rostami et al., 2024), its performance is equal to mT0XL, a multilingual 3.7B LLM.
- The performance gap between the best-performing model, GPT-4, and human averages around 27%. In subjects like mathematics, this gap widens to 31% accuracy for GPT-4. This exhibits a real challenge of current LLMs in solving human-level questions, especially in complex mathematical questions that need high-level mathematical calculation and reasoning skills.
- The models exhibit weaker performance in mathematics and natural science main categories compared to humanities and social sciences. This indicates their weaker performance on questions requiring high reasoning skills, compared to those mainly reliant on models' inherent knowledge. This underscores the necessity for enhancing the models' reasoning ability in the Persian language.
- The CoT has improved the performance of GPT-3.5 in Mathematics questions by 10%, but decreased the performance in Humanities questions by 8%.

## 5.2   Accuracy trends

We demonstrated the accuracy of the models on all three difficulty levels of questions for different educational stages and question publication years in Figure 9 and Figure 10 in the Appendix E, respectively. Key findings from these figures include:

- Most model accuracies decline with the increasing publication year for questions with medium and easy difficulty levels, while human performance remains consistent. This suggests that humans may have adapted to the evolving questions' difficulty over time, whereas models have not.
- There exists a notable performance gap between difficult and medium questions in human performance, indicating that difficulty level has a stronger impact on human performance compared to LLMs. Since humans assigned difficulty levels to questions, this gap may stem from differing perceptions of difficulty between humans and LLMs.
- GPT-4 outperforms humans in difficult questions within four years, as well as during the initial two educational stages (LPS and UPS). This result suggests that modern LLMs may excel at analyzing difficult questions compared to humans.

### 5.3 Selected choice distribution

Figures 12, 13, 14 in the Appendix E depicts the selected choice distributions of various models. It reveals that despite GPT-3.5 having the lowest count of unanswered questions, GPT-4 surpasses it in accuracy, indicating a more refined understanding despite its higher non-response rate. Comparatively, GPT-4 exhibits a more uniform distribution of choice selection, aligning closely with the Ground Truth and showing less bias than its predecessor, GPT-3.5, which tends towards selecting the second and third choices. High-performance models like GPT-4 demonstrate a closer alignment with Ground Truth distribution, indicating a lower bias level and potentially higher utility in applications.

### 5.4 Trap analysis

Table 16 in the Appendix F compares the performance of models and humans on trapped questions, using an x/y format where x represents overall accuracy and y the accuracy on trapped questions. The data shows that while traps often mislead students, leading to nearly random performance, models like GPT-4 exhibit only a slight drop in accuracy when faced with these traps. However, this drop in accuracy is most pronounced in the Social Sciences and Humanities. Notably, GPT-4 outperforms humans across all main categories in handling trapped questions, indicating its robustness against misleading choices and affirming the different perspectives of difficulty between humans and AI models.

### 5.5 Difficulty levels analysis

To explore how humans and LLMs perceive the difficulty of questions, we examined the accuracy of different models across three difficulty levels. Although our dataset included five labels ranging from easy to difficult to denote question difficulty, we combined the two relatively easy and relatively difficult labels into "easy" and "difficult", respectively, to ensure a more balanced distribution of question difficulty. Our experiment revealed a consistent trend: as question difficulty increased, both human and model answering accuracy decreased (See Tables 13, 14, 15 in the Appendix E).

Notably, in analytical and knowledge-based topics such as the humanities category, the GPT-4 model demonstrated superior performance compared to humans in tackling difficult questions. Conversely, models exhibiting more human-like performance, such as GPT-4, revealed that humans significantly outperformed them on easier questions. For additional statistical results, refer to Appendix G.

## 6 Conclusions

We introduced **Khayyam Challenge**, also known as **PersianMMLU**, as the first framework for assessing LLMs in the Persian language across various tasks, difficulty levels, and educational stages. This framework includes comprehensive metadata such as human performance, difficulty levels, and traps. Our assessment encompassed examining the performance of current LLMs on these datasets, evaluating their ability to extract answers (probabilistic and rule-based paradigms), and considering various aspects highlighted in the metadata. Our findings revealed that while LLMs demonstrated relatively satisfactory performance in question-solving tasks (especially on GPT-4), they still significantly lagged behind human performance, particularly in tasks necessitating reasoning. Moreover, analysis of metadata concerning difficulty levels and trapped questions unveiled notable discrepancies between model and human behavior, suggesting fundamental differences in learning approaches. This underscores the necessity for adaptations in LLMs training methodologies to achieve human-like proficiency. For future works, we aim to develop an LLM that bridges the performance gap between existing open-source models and GPT-4.

## 7 Acknowledgement

The authors would like to express their sincere gratitude to Pellekan Yadgiri (Learning Ladder), a sub-organization of Kanoon Farhangi Amoozesh (Cultural Educational Center), for providing the valuable dataset and related metadata that was crucial for this research. We are particularly grateful to Masoud Tajfard for his kind assistance in facilitating the acquisition of this data, which was permitted for research purposes. The dataset and metadata made available by Pellekan Yadgiri were instrumental in enabling the analyses and findings presented in this work.

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

# A  Data

| | |
|---|---|
| **Question** | شنونده‌ای که مساحت پرده گوشش ۶۰ میلی‌متر مربع است، تراز شدت صوت حاصل از یک منبع را ۵۰ دسی‌بل ، احساس می‌کند انرژی که در مدت ۵۰ ثانیه به پرده گوش این شنونده می‌رسد، چند میکروژول است؟ |
| **Correct Translation** | A listener with an eardrum area of $60mm^2$ perceives the sound intensity level from a source as $50$ decibels. How much energy reaches the listener's ear in 50 seconds in microjoules? |
| **API Translation** | A listener with an ear canal area of $60mm^2$ perceives the sound intensity level from a source as $50$ decibels. How much energy reaches the listener's ear in 50 seconds? |
| **Question** | در بیت زیر، شاعر از کدام اختیارات شاعری بهره برده است؟ «گویی بطّ سفید جامه به صابون زده است / کبک دری ساق پای در قدح خون زده است» |
| **Correct Translation** | In the following verse, which poetic devices has the poet employed? "The duck's feathers are so white, as if they've been washed with soap / The partridge's legs are so red, as if they've been dipped in a cup of blood" |
| **API Translation** | In the following verse, which poetic options did the poet use? "It's as if a white duck has put its clothes on soap / the partridge has put its leg in a glass of blood" |
| **Question** | کیان پسر عمه زهرا، پسر خاله من است. من ۲ خاله و ۳ دایی دارم. زهرا به‌ترتیب چند عمه و چند عمو دارد؟ |
| **Correct Translation** | Kian is the son of the sister of Zahra's Father, and he is the son of my mother's sister. My mother has two sisters and three brothers. How many sisters and brothers does Zahra's father have? |
| **API Translation** | Kian is the son of my aunt Zahra. I have 2 aunts and 3 uncles. How many aunts and how many uncles does Zahra have respectively? |

Table 4: Examples illustrating information loss in domain-specific translations from English to Persian Using the GPT-3.5 API.

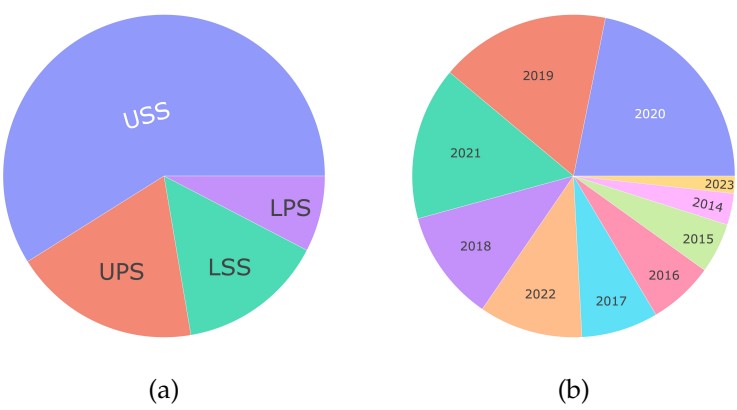

(a)                              (b)

Figure 2: Distribution of questions across publication year and educational stage

| Main Category | Easy | Medium | Difficult | Sum |
|---|---|---|---|---|
| Social Science | 678 | 1552 | 1194 | 3424 |
| Mathematics | 1461 | 3303 | 2300 | 7064 |
| Natural Science | 1299 | 2554 | 1999 | 5852 |
| Humanities | 724 | 1900 | 1841 | 4465 |
| Sum | 4162 | 9309 | 7334 | 20805 |

Table 5: Question distribution across main categories by difficulty level

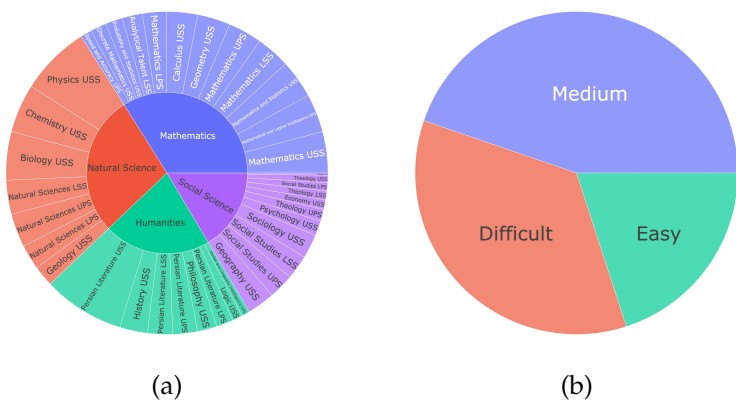

Figure 3: Question distribution with respect to categories, level of difficulty, and educational stages. LPS: Lower Primary School, UPS: Upper Primary School, LSS: Lower Secondary School, USS: Upper Secondary School

## B Models

**GPT-4**

OpenAI has introduced GPT-4 as a large multimodal model. However, the internal structure and operational specifics of GPT-4 are proprietary and not openly disclosed.

**GPT-3.5**

GPT-3.5, created by OpenAI, represents a significant advancement in natural language processing technology. It boasts enhanced capabilities in understanding and generating human-like text, performing impressively across various language tasks like text completion, translation, and question answering, thanks to its expansive architecture and vast training data.

**Claude3-haiku**

The Claude 3 Haiku model, developed recently by Anthropic, are proprietary models with undisclosed architecture and training specifics.

**Aya**

Aya, an open-source model by Cohere, is a massively multilingual generative language model capable of understanding instructions in 101 languages. Over half of these languages are classified as lower-resourced.

**XVERSE-13B**

XVERSE-13B, created by Shenzhen Yuanxiang Technology, stands as a versatile multilingual large language model. Employing a prominent Decoder-only Transformer network structure, XVERSE-13B has a context length of 8k. It has been meticulously trained on a vast and varied dataset containing 1.4 trillion tokens, covering more than 40 languages including Chinese, English, Russian, and Spanish.

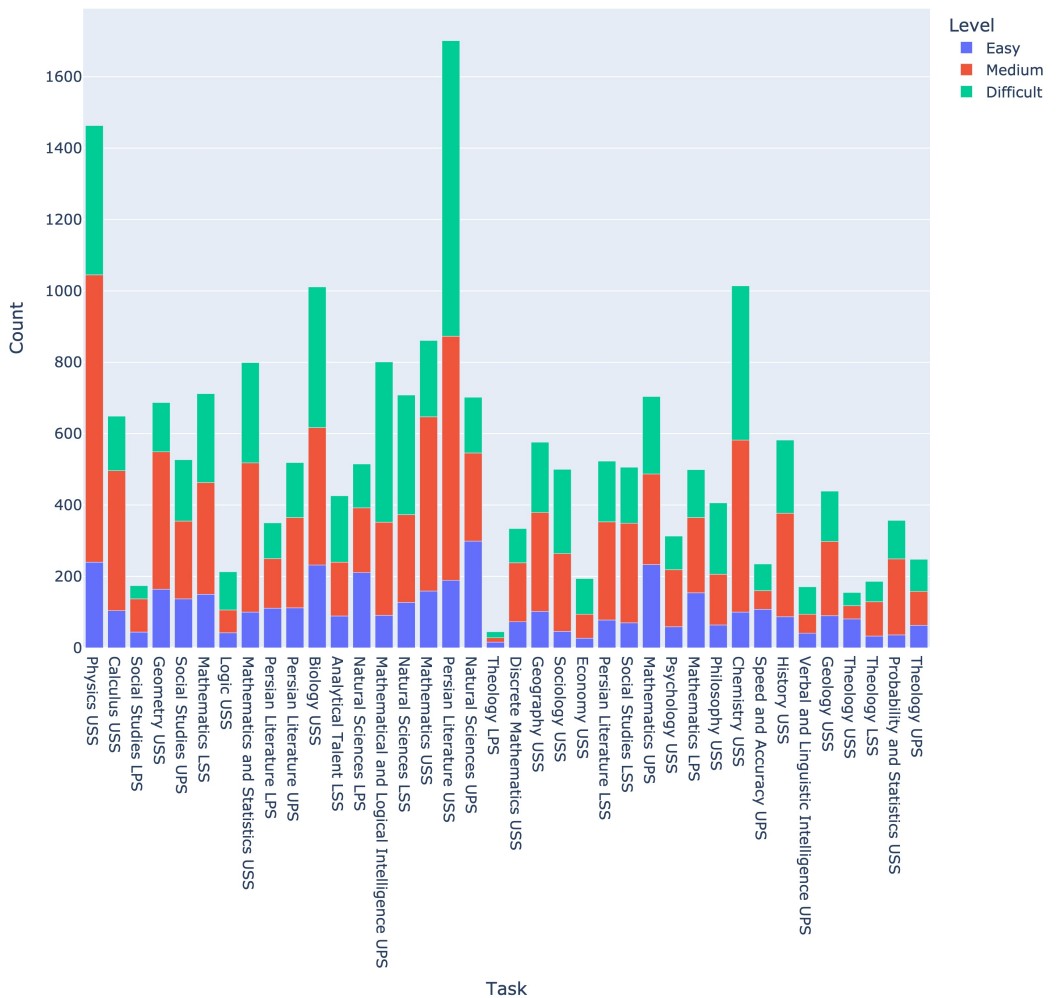

Figure 4: Question distribution across all tasks by difficulty level

**XVERSE-7B**

XVERSE-7B, much like its counterpart XVERSE-13B, is a multilingual large language model developed by Shenzhen Yuanxiang Technology. It shares the same mainstream Decoder-only Transformer network structure but with a slightly reduced parameter size of 7 billion.

**PersianMind**

PersianMind is an openly available bilingual large language model. It enriches LLaMa2's vocabulary by integrating 10,000 Persian tokens and undergoes training on a dataset containing nearly 2 billion Persian tokens.

**mT0XL**

mT0XL originates from the fine-tuning of BLOOM models using xP3, a dataset that encompasses multilingual content coupled with English prompts. The BLOOM models, which are extensive decoder-only language models, have been pre-trained on roughly 350 billion tokens. They share structural similarities with GPT-3.

**mGPT**

mGPT is an open-source adaptation of GPT-3, having multilingual capabilities. It undergoes pretraining across 61 languages, drawn from 25 diverse language families, leveraging Wikipedia and the C4 Corpus. Additionally, mGPT offers support for Persian language integration.

## C   Prompt template

در زیر سوالات چند گزینه‌ای (با پاسخ) در مورد   هندسه دوره دوم متوسطه   را مشاهده می‌کنید.

سوال:

به ازای هر $m$، معادله $(m-\Upsilon)x + (m+\Upsilon)y = \mathcal{F}$، معادله قطری از دایره $C$ است. اگر نقطه $A(-\Upsilon, \Upsilon)$ روی دایره $C$ باشد، محیط دایره $C$ کدام است؟

گزینه‌ها:
$\Upsilon\sqrt{\Upsilon}\pi$ (۱
$\Upsilon\pi$ (۲
$\Upsilon\pi$ (۳
$\Upsilon\sqrt{\Upsilon}\pi$ (۴
جواب:

Below you will find multiple-choice questions (with answers) regarding **geometry for upper secondary school**.

Question: **For each $m$ the equation $(m-2)x + (m+1)y = 6$, is the equation of chord of a circle $C$. If the point $A(-1, 1)$ lies on the circle $C$, what is the circumference of circle $C$?**

Choices:
1) $2\sqrt{2}\pi$
2) $2\pi$
3) $3\pi$
4) $2\sqrt{3}\pi$

Answer:

Figure 5: Sample prompt-0 with English translation for enhanced readability

در زیر سوالات چند گزینه‌ای (با پاسخ) در مورد   هندسه دوره دوم متوسطه   را مشاهده می‌کنید.

سوال:

به ازای هر $m$، معادله $\mathfrak{F} = y(\mathfrak{1} + m) + x(\mathfrak{1} - m)$، معادله قطری از دایره $C$ است. اگر نقطه   $A(-\mathfrak{1}, \mathfrak{1})$ روی دایره $C$ باشد، محیط دایره $C$ کدام است؟

گزینه‌ها:

a) $2\sqrt{2}\pi$
b) $2\pi$
c) $3\pi$
d) $2\sqrt{3}\pi$

جواب:

Below you will find multiple-choice questions (with answers) regarding **geometry for upper secondary school**.

Question: **For each $m$ the equation $(m-2)x + (m+1)y = 6$, is the equation of chord of a circle $C$. If the point $A(-1,1)$ lies on the circle $C$, what is the circumference of circle $C$?**

Choices:

a) $2\sqrt{2}\pi$
b) $2\pi$
c) $3\pi$
d) $2\sqrt{3}\pi$

Answer:

Figure 6: Sample prompt-1 with English translation for enhanced readability

The following are multiple choice questions (with answer) about Upper Secondary School Geometry

سوال:

به ازای هر $m$، معادله $(m - ۲)x + (m + ۱)y = ۶$، معادله قطری از دایره $C$ است. اگر نقطه $A(-۱, ۱)$ روی دایره $C$ باشد، محیط دایره $C$ کدام است؟

گزینه‌ها:

  a) $2\sqrt{2}\pi$
  b) $2\pi$
  c) $3\pi$
  d) $2\sqrt{3}\pi$

جواب:

Below you will find multiple-choice questions (with answers) regarding **geometry for upper secondary school**.

Question: **For each $m$ the equation $(m - 2)x + (m + 1)y = 6$, is the equation of chord of a circle $C$. If the point $A(-1, 1)$ lies on the circle $C$, what is the circumference of circle $C$?**

Choices:

  a) $2\sqrt{2}\pi$
  b) $2\pi$
  c) $3\pi$
  d) $2\sqrt{3}\pi$

Answer:

Figure 7: Sample prompt-2 with English translation for enhanced readability

در زیر سوالات چند گزینه‌ای (با پاسخ) در مورد   هندسه دوره دوم متوسطه   را مشاهده می‌کنید.

سوال:

به ازای هر $m$، معادله $۶ = (m + ۱)y + (m - ۲)x$، معادله قطری از دایره $C$ است. اگر نقطه   $A(-۱, ۱)$ روی دایره $C$ باشد، محیط دایره $C$ کدام است؟

گزینه‌ها:

a) $2\sqrt{2}\pi$
b) $2\pi$
c) $3\pi$
d) $2\sqrt{3}\pi$

جواب: بیایید قدم به قدم فکر کنیم

Below you will find multiple-choice questions (with answers) regarding **geometry for upper secondary school**.

Question: **For each $m$ the equation $(m - 2)x + (m + 1)y = 6$, is the equation of chord of a circle $C$. If the point $A(-1, 1)$ lies on the circle $C$, what is the circumference of circle $C$?**

Choices:

a) $2\sqrt{2}\pi$
b) $2\pi$
c) $3\pi$
d) $2\sqrt{3}\pi$

Answer: Let's think step by step

Figure 8: Sample CoT prompt with English translation for enhanced readability

## D   Prompt & CoT

| Main Categories | Prompt-0 | Prompt-1 | Prompt-2 |
|---|---|---|---|
| Humanities | 0.32 | 0.35 | 0.36 |
| Mathematics | 0.28 | 0.3 | 0.34 |
| Natural Science | 0.35 | 0.36 | 0.37 |
| Social Science | 0.43 | 0.42 | 0.42 |
| Avg on all tasks | 0.34 | 0.36 | 0.37 |
| Avg on all questions | 0.35 | 0.36 | 0.37 |

Table 6: Accuracy of GPT-3.5 across main categories for different prompts

| Main Categories | Prompt-0 | CoT |
|---|---|---|
| Humanities | 0.36 | 0.28 |
| Mathematics | 0.31 | 0.41 |
| Avg on all tasks | 0.33 | 0.34 |
| Avg on all questions | 0.31 | 0.4 |

Table 7: Accuracy of GPT-3.5 across main categories for main prompt and CoT

# E Accuracy & candidate choice distribution

| Main Categories | Human | GPT-4 | Haiku | GPT-3.5 | Aya | XVERSE-13B | XVERSE-7B | PersianMind | mT0XL | mGPT | Random |
|---|---|---|---|---|---|---|---|---|---|---|---|
| Humanities | 0.71 | 0.47 | 0.43 | 0.33 | 0.27 | 0.3 | 0.27 | 0.25 | 0.26 | 0.24 | 0.25 |
| Mathematics | 0.72 | 0.41 | 0.33 | 0.32 | 0.24 | 0.26 | 0.26 | 0.25 | 0.24 | 0.23 | 0.25 |
| Natural Science | 0.79 | 0.49 | 0.42 | 0.37 | 0.29 | 0.31 | 0.29 | 0.25 | 0.27 | 0.24 | 0.25 |
| Social Science | 0.85 | 0.64 | 0.57 | 0.44 | 0.34 | 0.35 | 0.33 | 0.28 | 0.29 | 0.25 | 0.25 |
| Avg on all tasks | 0.77 | 0.5 | 0.44 | 0.37 | 0.28 | 0.3 | 0.29 | 0.26 | 0.26 | 0.24 | 0.25 |
| Avg on all questions | 0.76 | 0.49 | 0.41 | 0.36 | 0.28 | 0.29 | 0.28 | 0.26 | 0.26 | 0.24 | 0.25 |

Table 8: Human vs model accuracy for main categories: model answers extracted via "Full Answer Probability" method

| Main Categories | Human | GPT-4 | Haiku | GPT-3.5 | Aya | XVERSE-13B | XVERSE-7B | PersianMind | mT0XL | mGPT | Random |
|---|---|---|---|---|---|---|---|---|---|---|---|
| Humanities | 0.71 | 0.47 | 0.43 | 0.33 | 0.32 | 0.29 | 0.27 | 0.26 | 0.27 | 0.23 | 0.25 |
| Mathematics | 0.72 | 0.41 | 0.33 | 0.32 | 0.27 | 0.26 | 0.27 | 0.25 | 0.27 | 0.23 | 0.25 |
| Natural Science | 0.79 | 0.49 | 0.42 | 0.37 | 0.33 | 0.31 | 0.29 | 0.27 | 0.28 | 0.21 | 0.25 |
| Social Science | 0.85 | 0.64 | 0.57 | 0.44 | 0.42 | 0.35 | 0.33 | 0.3 | 0.32 | 0.22 | 0.25 |
| Avg on all tasks | 0.77 | 0.5 | 0.44 | 0.37 | 0.34 | 0.3 | 0.29 | 0.27 | 0.28 | 0.22 | 0.25 |
| Avg on all questions | 0.76 | 0.49 | 0.41 | 0.36 | 0.32 | 0.29 | 0.28 | 0.27 | 0.28 | 0.22 | 0.25 |

Table 9: Human vs model accuracy for main categories: model answers extracted via "Single Token Probability" method

| Task | Human | GPT-4 | Haiku | GPT-3.5 | Aya | XVERSE-13B | XVERSE-7B | PersianMind | mT0XL | mGPT | Random |
|---|---|---|---|---|---|---|---|---|---|---|---|
| Analytical Talent LSS | 0.59 | 0.52 | 0.38 | 0.35 | 0.29 | 0.3 | 0.31 | 0.26 | 0.28 | 0.26 | 0.25 |
| Calculus USS | 0.84 | 0.32 | 0.29 | 0.26 | 0.26 | 0.24 | 0.26 | 0.22 | 0.25 | 0.23 | 0.25 |
| Chemistry USS | 0.73 | 0.37 | 0.31 | 0.31 | 0.3 | 0.28 | 0.27 | 0.28 | 0.27 | 0.25 | 0.26 |
| Discrete Mathematics USS | 0.77 | 0.32 | 0.25 | 0.25 | 0.24 | 0.26 | 0.26 | 0.27 | 0.29 | 0.27 | 0.24 |
| Economy USS | 0.69 | 0.63 | 0.51 | 0.4 | 0.47 | 0.33 | 0.25 | 0.28 | 0.3 | 0.22 | 0.28 |
| Geography USS | 0.85 | 0.56 | 0.5 | 0.38 | 0.36 | 0.31 | 0.31 | 0.31 | 0.29 | 0.27 | 0.24 |
| Geology USS | 0.89 | 0.58 | 0.53 | 0.4 | 0.35 | 0.32 | 0.24 | 0.28 | 0.25 | 0.26 | 0.25 |
| Geometry USS | 0.83 | 0.34 | 0.31 | 0.3 | 0.28 | 0.25 | 0.28 | 0.28 | 0.26 | 0.23 | 0.28 |
| History USS | 0.86 | 0.56 | 0.47 | 0.36 | 0.32 | 0.33 | 0.29 | 0.31 | 0.27 | 0.23 | 0.26 |
| Logic USS | 0.58 | 0.42 | 0.4 | 0.31 | 0.28 | 0.33 | 0.29 | 0.31 | 0.23 | 0.28 | 0.23 |
| Mathematical and Logical Intelligence UPS | 0.48 | 0.43 | 0.3 | 0.32 | 0.29 | 0.27 | 0.31 | 0.26 | 0.27 | 0.28 | 0.25 |
| Mathematics LPS | 0.78 | 0.56 | 0.37 | 0.37 | 0.3 | 0.32 | 0.32 | 0.23 | 0.26 | 0.26 | 0.26 |
| Mathematics LSS | 0.75 | 0.4 | 0.36 | 0.32 | 0.32 | 0.3 | 0.27 | 0.28 | 0.24 | 0.28 | 0.24 |
| Mathematics UPS | 0.68 | 0.49 | 0.37 | 0.35 | 0.27 | 0.27 | 0.26 | 0.27 | 0.27 | 0.25 | 0.23 |
| Mathematics USS | 0.85 | 0.34 | 0.3 | 0.31 | 0.27 | 0.26 | 0.26 | 0.26 | 0.25 | 0.25 | 0.25 |
| Mathematics and Statistics USS | 0.61 | 0.43 | 0.37 | 0.34 | 0.27 | 0.28 | 0.28 | 0.28 | 0.28 | 0.26 | 0.22 |
| Natural Sciences LPS | 0.85 | 0.81 | 0.7 | 0.61 | 0.5 | 0.47 | 0.41 | 0.34 | 0.38 | 0.26 | 0.28 |
| Natural Sciences LSS | 0.71 | 0.59 | 0.46 | 0.39 | 0.33 | 0.32 | 0.3 | 0.3 | 0.28 | 0.27 | 0.22 |
| Natural Sciences UPS | 0.83 | 0.73 | 0.62 | 0.48 | 0.42 | 0.39 | 0.32 | 0.3 | 0.29 | 0.24 | 0.26 |
| Persian Literature LPS | 0.84 | 0.66 | 0.55 | 0.42 | 0.4 | 0.39 | 0.3 | 0.29 | 0.33 | 0.23 | 0.22 |
| Persian Literature LSS | 0.77 | 0.51 | 0.46 | 0.34 | 0.29 | 0.29 | 0.28 | 0.26 | 0.29 | 0.27 | 0.27 |
| Persian Literature UPS | 0.79 | 0.57 | 0.53 | 0.38 | 0.32 | 0.35 | 0.28 | 0.25 | 0.31 | 0.24 | 0.25 |
| Persian Literature USS | 0.59 | 0.35 | 0.32 | 0.28 | 0.29 | 0.25 | 0.26 | 0.26 | 0.26 | 0.25 | 0.25 |
| Philosophy USS | 0.61 | 0.53 | 0.51 | 0.39 | 0.36 | 0.31 | 0.26 | 0.31 | 0.33 | 0.26 | 0.22 |
| Physics USS | 0.83 | 0.39 | 0.32 | 0.28 | 0.27 | 0.28 | 0.25 | 0.27 | 0.27 | 0.24 | 0.25 |
| Probability and Statistics USS | 0.78 | 0.42 | 0.37 | 0.32 | 0.25 | 0.23 | 0.25 | 0.22 | 0.23 | 0.2 | 0.28 |
| Psychology USS | 0.78 | 0.63 | 0.5 | 0.4 | 0.4 | 0.32 | 0.35 | 0.32 | 0.35 | 0.21 | 0.27 |
| Social Studies LPS | 0.94 | 0.85 | 0.76 | 0.67 | 0.62 | 0.59 | 0.44 | 0.46 | 0.37 | 0.3 | 0.27 |
| Social Studies LSS | 0.86 | 0.69 | 0.57 | 0.43 | 0.39 | 0.35 | 0.34 | 0.29 | 0.3 | 0.22 | 0.26 |
| Social Studies UPS | 0.89 | 0.73 | 0.7 | 0.52 | 0.46 | 0.35 | 0.36 | 0.38 | 0.37 | 0.24 | 0.22 |
| Sociology USS | 0.82 | 0.49 | 0.44 | 0.34 | 0.37 | 0.32 | 0.27 | 0.28 | 0.26 | 0.26 | 0.25 |
| Speed and Accuracy UPS | 0.85 | 0.41 | 0.31 | 0.38 | 0.26 | 0.29 | 0.27 | 0.25 | 0.28 | 0.27 | 0.24 |
| Theology LPS | 0.81 | 0.82 | 0.73 | 0.67 | 0.53 | 0.38 | 0.38 | 0.47 | 0.56 | 0.24 | 0.27 |
| Theology LSS | 0.87 | 0.62 | 0.6 | 0.49 | 0.46 | 0.39 | 0.34 | 0.32 | 0.38 | 0.28 | 0.2 |
| Theology UPS | 0.91 | 0.8 | 0.7 | 0.57 | 0.48 | 0.44 | 0.37 | 0.32 | 0.37 | 0.19 | 0.24 |
| Theology USS | 0.8 | 0.54 | 0.44 | 0.33 | 0.37 | 0.26 | 0.26 | 0.24 | 0.34 | 0.2 | 0.26 |
| Verbal and Linguistic Intelligence UPS | 0.62 | 0.54 | 0.49 | 0.42 | 0.35 | 0.37 | 0.26 | 0.28 | 0.3 | 0.26 | 0.25 |
| Biology USS | 0.73 | 0.32 | 0.28 | 0.31 | 0.3 | 0.28 | 0.28 | 0.3 | 0.27 | 0.27 | 0.27 |
| Avg on all tasks | 0.77 | 0.53 | 0.46 | 0.39 | 0.35 | 0.32 | 0.3 | 0.29 | 0.3 | 0.25 | 0.25 |
| Avg on all questions | 0.76 | 0.49 | 0.41 | 0.36 | 0.32 | 0.3 | 0.29 | 0.28 | 0.28 | 0.25 | 0.25 |

Table 10: Accuracy of humans and different models across all subjects and educational stages: model answers extracted via "Regex" method. LPS: Lower Primary School, UPS: Upper Primary School, LSS: Lower Secondary School, USS: Upper Secondary School

| Task | Human | GPT-4 | Haiku | GPT-3.5 | Aya | XVERSE-13B | XVERSE-7B | PersianMind | mGPT | mT0XL | Random |
|---|---|---|---|---|---|---|---|---|---|---|---|
| Analytical Talent LSS | 0.59 | 0.52 | 0.38 | 0.35 | 0.27 | 0.3 | 0.32 | 0.27 | 0.22 | 0.29 | 0.25 |
| Calculus USS | 0.84 | 0.32 | 0.29 | 0.26 | 0.26 | 0.27 | 0.3 | 0.24 | 0.25 | 0.26 | 0.25 |
| Chemistry USS | 0.73 | 0.37 | 0.31 | 0.31 | 0.3 | 0.28 | 0.27 | 0.24 | 0.23 | 0.28 | 0.26 |
| Discrete Mathematics USS | 0.77 | 0.32 | 0.25 | 0.25 | 0.24 | 0.28 | 0.28 | 0.28 | 0.25 | 0.27 | 0.24 |
| Economy USS | 0.69 | 0.63 | 0.51 | 0.4 | 0.43 | 0.32 | 0.24 | 0.3 | 0.21 | 0.26 | 0.28 |
| Geography USS | 0.85 | 0.56 | 0.5 | 0.38 | 0.36 | 0.31 | 0.3 | 0.3 | 0.24 | 0.28 | 0.24 |
| Geology USS | 0.89 | 0.58 | 0.53 | 0.4 | 0.35 | 0.31 | 0.24 | 0.21 | 0.2 | 0.26 | 0.25 |
| Geometry USS | 0.83 | 0.34 | 0.31 | 0.3 | 0.28 | 0.26 | 0.27 | 0.25 | 0.24 | 0.26 | 0.28 |
| History USS | 0.86 | 0.56 | 0.47 | 0.36 | 0.33 | 0.3 | 0.29 | 0.28 | 0.21 | 0.27 | 0.26 |
| Logic USS | 0.58 | 0.42 | 0.4 | 0.31 | 0.29 | 0.3 | 0.28 | 0.29 | 0.29 | 0.28 | 0.23 |
| Mathematical and Logical Intelligence UPS | 0.48 | 0.43 | 0.3 | 0.32 | 0.28 | 0.27 | 0.29 | 0.26 | 0.23 | 0.27 | 0.25 |
| Mathematics LPS | 0.78 | 0.56 | 0.37 | 0.37 | 0.29 | 0.24 | 0.27 | 0.27 | 0.22 | 0.27 | 0.26 |
| Mathematics LSS | 0.75 | 0.4 | 0.36 | 0.32 | 0.31 | 0.23 | 0.25 | 0.25 | 0.21 | 0.26 | 0.24 |
| Mathematics UPS | 0.68 | 0.49 | 0.37 | 0.35 | 0.26 | 0.25 | 0.26 | 0.24 | 0.23 | 0.24 | 0.23 |
| Mathematics USS | 0.85 | 0.34 | 0.3 | 0.31 | 0.27 | 0.24 | 0.24 | 0.25 | 0.22 | 0.28 | 0.25 |
| Mathematics and Statistics USS | 0.61 | 0.43 | 0.37 | 0.34 | 0.26 | 0.3 | 0.27 | 0.27 | 0.24 | 0.28 | 0.22 |
| Natural Sciences LPS | 0.85 | 0.81 | 0.7 | 0.61 | 0.5 | 0.44 | 0.41 | 0.35 | 0.19 | 0.37 | 0.28 |
| Natural Sciences LSS | 0.71 | 0.59 | 0.46 | 0.39 | 0.34 | 0.31 | 0.29 | 0.27 | 0.18 | 0.3 | 0.22 |
| Natural Sciences UPS | 0.83 | 0.73 | 0.62 | 0.48 | 0.42 | 0.36 | 0.31 | 0.29 | 0.18 | 0.28 | 0.26 |
| Persian Literature LPS | 0.84 | 0.66 | 0.55 | 0.42 | 0.41 | 0.37 | 0.3 | 0.26 | 0.19 | 0.33 | 0.22 |
| Persian Literature LSS | 0.77 | 0.51 | 0.46 | 0.34 | 0.32 | 0.3 | 0.28 | 0.29 | 0.27 | 0.26 | 0.27 |
| Persian Literature UPS | 0.79 | 0.57 | 0.53 | 0.38 | 0.35 | 0.32 | 0.28 | 0.25 | 0.2 | 0.31 | 0.25 |
| Persian Literature USS | 0.59 | 0.35 | 0.32 | 0.28 | 0.28 | 0.25 | 0.26 | 0.25 | 0.23 | 0.25 | 0.25 |
| Philosophy USS | 0.61 | 0.53 | 0.51 | 0.39 | 0.34 | 0.31 | 0.26 | 0.29 | 0.27 | 0.26 | 0.22 |
| Physics USS | 0.83 | 0.39 | 0.32 | 0.28 | 0.27 | 0.27 | 0.25 | 0.26 | 0.22 | 0.28 | 0.25 |
| Probability and Statistics USS | 0.78 | 0.42 | 0.37 | 0.32 | 0.25 | 0.25 | 0.22 | 0.23 | 0.2 | 0.26 | 0.28 |
| Psychology USS | 0.78 | 0.63 | 0.5 | 0.4 | 0.38 | 0.35 | 0.36 | 0.31 | 0.24 | 0.33 | 0.27 |
| Social Studies LPS | 0.94 | 0.85 | 0.76 | 0.67 | 0.6 | 0.53 | 0.43 | 0.36 | 0.17 | 0.32 | 0.27 |
| Social Studies LSS | 0.86 | 0.69 | 0.57 | 0.43 | 0.4 | 0.35 | 0.34 | 0.29 | 0.22 | 0.32 | 0.26 |
| Social Studies UPS | 0.89 | 0.73 | 0.7 | 0.52 | 0.48 | 0.36 | 0.37 | 0.34 | 0.2 | 0.32 | 0.22 |
| Sociology USS | 0.82 | 0.49 | 0.44 | 0.34 | 0.37 | 0.3 | 0.27 | 0.24 | 0.22 | 0.28 | 0.25 |
| Speed and Accuracy UPS | 0.85 | 0.41 | 0.31 | 0.38 | 0.25 | 0.24 | 0.26 | 0.25 | 0.22 | 0.26 | 0.24 |
| Theology LPS | 0.81 | 0.82 | 0.73 | 0.67 | 0.58 | 0.33 | 0.38 | 0.36 | 0.11 | 0.44 | 0.27 |
| Theology LSS | 0.87 | 0.62 | 0.6 | 0.49 | 0.46 | 0.36 | 0.32 | 0.35 | 0.23 | 0.37 | 0.2 |
| Theology UPS | 0.91 | 0.8 | 0.7 | 0.57 | 0.46 | 0.43 | 0.36 | 0.33 | 0.2 | 0.43 | 0.24 |
| Theology USS | 0.8 | 0.54 | 0.44 | 0.33 | 0.33 | 0.25 | 0.26 | 0.22 | 0.22 | 0.27 | 0.26 |
| Verbal and Linguistic Intelligence UPS | 0.62 | 0.54 | 0.49 | 0.42 | 0.42 | 0.36 | 0.26 | 0.25 | 0.22 | 0.33 | 0.25 |
| Biology USS | 0.73 | 0.32 | 0.28 | 0.31 | 0.29 | 0.28 | 0.3 | 0.25 | 0.23 | 0.25 | 0.27 |
| Avg on all tasks | 0.77 | 0.53 | 0.46 | 0.39 | 0.35 | 0.31 | 0.29 | 0.28 | 0.22 | 0.29 | 0.25 |
| Avg on all questions | 0.76 | 0.49 | 0.41 | 0.36 | 0.32 | 0.29 | 0.28 | 0.27 | 0.22 | 0.28 | 0.25 |

Table 11: Accuracy of human and different models across all subjects and educational stages: model answers extracted via "Single Token Probability" method. LPS: Lower Primary School, UPS: Upper Primary School, LSS: Lower Secondary School, USS: Upper Secondary School

| Task | Human | GPT-4 | Haiku | GPT-3.5 | Aya | XVERSE-13B | XVERSE-7B | PersianMind | mT0XL | mGPT | Random |
|---|---|---|---|---|---|---|---|---|---|---|---|
| Analytical Talent LSS | 0.59 | 0.52 | 0.38 | 0.35 | 0.28 | 0.3 | 0.32 | 0.29 | 0.26 | 0.23 | 0.25 |
| Calculus USS | 0.84 | 0.32 | 0.29 | 0.26 | 0.22 | 0.24 | 0.28 | 0.23 | 0.22 | 0.22 | 0.25 |
| Chemistry USS | 0.73 | 0.37 | 0.31 | 0.31 | 0.27 | 0.28 | 0.28 | 0.26 | 0.26 | 0.23 | 0.26 |
| Discrete Mathematics USS | 0.77 | 0.32 | 0.25 | 0.25 | 0.23 | 0.28 | 0.28 | 0.28 | 0.28 | 0.25 | 0.24 |
| Economy USS | 0.69 | 0.63 | 0.51 | 0.4 | 0.27 | 0.34 | 0.23 | 0.25 | 0.23 | 0.23 | 0.28 |
| Geography USS | 0.85 | 0.56 | 0.5 | 0.38 | 0.31 | 0.32 | 0.3 | 0.29 | 0.26 | 0.26 | 0.24 |
| Geology USS | 0.89 | 0.58 | 0.53 | 0.4 | 0.29 | 0.31 | 0.23 | 0.27 | 0.25 | 0.23 | 0.25 |
| Geometry USS | 0.83 | 0.34 | 0.31 | 0.3 | 0.24 | 0.24 | 0.25 | 0.23 | 0.23 | 0.24 | 0.28 |
| History USS | 0.86 | 0.56 | 0.47 | 0.36 | 0.26 | 0.3 | 0.29 | 0.26 | 0.25 | 0.23 | 0.26 |
| Logic USS | 0.58 | 0.42 | 0.4 | 0.31 | 0.27 | 0.31 | 0.29 | 0.23 | 0.22 | 0.25 | 0.23 |
| Mathematical and Logical Intelligence UPS | 0.48 | 0.43 | 0.3 | 0.32 | 0.23 | 0.26 | 0.29 | 0.25 | 0.25 | 0.24 | 0.25 |
| Mathematics LPS | 0.78 | 0.56 | 0.37 | 0.37 | 0.24 | 0.25 | 0.27 | 0.25 | 0.25 | 0.25 | 0.26 |
| Mathematics LSS | 0.75 | 0.4 | 0.36 | 0.32 | 0.26 | 0.24 | 0.25 | 0.24 | 0.22 | 0.23 | 0.24 |
| Mathematics UPS | 0.68 | 0.49 | 0.37 | 0.35 | 0.25 | 0.24 | 0.25 | 0.27 | 0.22 | 0.23 | 0.23 |
| Mathematics USS | 0.85 | 0.34 | 0.3 | 0.31 | 0.22 | 0.23 | 0.23 | 0.23 | 0.23 | 0.23 | 0.25 |
| Mathematics and Statistics USS | 0.61 | 0.43 | 0.37 | 0.34 | 0.25 | 0.3 | 0.27 | 0.28 | 0.23 | 0.23 | 0.22 |
| Natural Sciences LPS | 0.85 | 0.81 | 0.7 | 0.61 | 0.38 | 0.44 | 0.41 | 0.25 | 0.29 | 0.22 | 0.28 |
| Natural Sciences LSS | 0.71 | 0.59 | 0.46 | 0.39 | 0.3 | 0.31 | 0.3 | 0.26 | 0.27 | 0.24 | 0.22 |
| Natural Sciences UPS | 0.83 | 0.73 | 0.62 | 0.48 | 0.34 | 0.37 | 0.32 | 0.27 | 0.25 | 0.23 | 0.26 |
| Persian Literature LPS | 0.84 | 0.66 | 0.55 | 0.42 | 0.32 | 0.37 | 0.3 | 0.25 | 0.24 | 0.22 | 0.22 |
| Persian Literature LSS | 0.77 | 0.51 | 0.46 | 0.34 | 0.26 | 0.3 | 0.28 | 0.23 | 0.26 | 0.24 | 0.27 |
| Persian Literature UPS | 0.79 | 0.57 | 0.53 | 0.38 | 0.27 | 0.32 | 0.28 | 0.26 | 0.28 | 0.25 | 0.25 |
| Persian Literature USS | 0.59 | 0.35 | 0.32 | 0.28 | 0.25 | 0.25 | 0.26 | 0.25 | 0.26 | 0.22 | 0.25 |
| Philosophy USS | 0.61 | 0.53 | 0.51 | 0.39 | 0.26 | 0.32 | 0.26 | 0.27 | 0.27 | 0.3 | 0.22 |
| Physics USS | 0.83 | 0.39 | 0.32 | 0.28 | 0.25 | 0.27 | 0.24 | 0.23 | 0.27 | 0.24 | 0.25 |
| Probability and Statistics USS | 0.78 | 0.42 | 0.37 | 0.32 | 0.23 | 0.24 | 0.22 | 0.23 | 0.28 | 0.19 | 0.28 |
| Psychology USS | 0.78 | 0.63 | 0.5 | 0.4 | 0.3 | 0.34 | 0.36 | 0.29 | 0.23 | 0.23 | 0.27 |
| Social Studies LPS | 0.94 | 0.85 | 0.76 | 0.67 | 0.47 | 0.53 | 0.44 | 0.33 | 0.29 | 0.25 | 0.27 |
| Social Studies LSS | 0.86 | 0.69 | 0.57 | 0.43 | 0.36 | 0.35 | 0.34 | 0.28 | 0.3 | 0.22 | 0.26 |
| Social Studies UPS | 0.89 | 0.73 | 0.7 | 0.52 | 0.38 | 0.36 | 0.37 | 0.28 | 0.32 | 0.26 | 0.22 |
| Sociology USS | 0.82 | 0.49 | 0.44 | 0.34 | 0.27 | 0.28 | 0.27 | 0.28 | 0.27 | 0.24 | 0.25 |
| Speed and Accuracy UPS | 0.85 | 0.41 | 0.31 | 0.38 | 0.27 | 0.25 | 0.29 | 0.24 | 0.26 | 0.24 | 0.24 |
| Theology LPS | 0.81 | 0.82 | 0.73 | 0.67 | 0.47 | 0.36 | 0.38 | 0.27 | 0.42 | 0.24 | 0.27 |
| Theology LSS | 0.87 | 0.62 | 0.6 | 0.49 | 0.36 | 0.38 | 0.32 | 0.26 | 0.35 | 0.29 | 0.2 |
| Theology UPS | 0.91 | 0.8 | 0.7 | 0.57 | 0.41 | 0.43 | 0.36 | 0.26 | 0.31 | 0.24 | 0.24 |
| Theology USS | 0.8 | 0.54 | 0.44 | 0.33 | 0.3 | 0.26 | 0.26 | 0.26 | 0.32 | 0.31 | 0.26 |
| Verbal and Linguistic Intelligence UPS | 0.62 | 0.54 | 0.49 | 0.42 | 0.35 | 0.39 | 0.25 | 0.29 | 0.3 | 0.27 | 0.25 |
| Biology USS | 0.73 | 0.32 | 0.28 | 0.31 | 0.28 | 0.29 | 0.29 | 0.26 | 0.25 | 0.28 | 0.27 |
| Avg on all tasks | 0.77 | 0.53 | 0.46 | 0.39 | 0.29 | 0.31 | 0.29 | 0.26 | 0.27 | 0.24 | 0.25 |
| Avg on all questions | 0.76 | 0.49 | 0.41 | 0.36 | 0.28 | 0.29 | 0.28 | 0.26 | 0.26 | 0.24 | 0.25 |

Table 12: Accuracy of human and different models across all subjects and educational stages: model answers extracted via "Full Answer Probability" method. LPS: Lower Primary School, UPS: Upper Primary School, LSS: Lower Secondary School, USS: Upper Secondary School

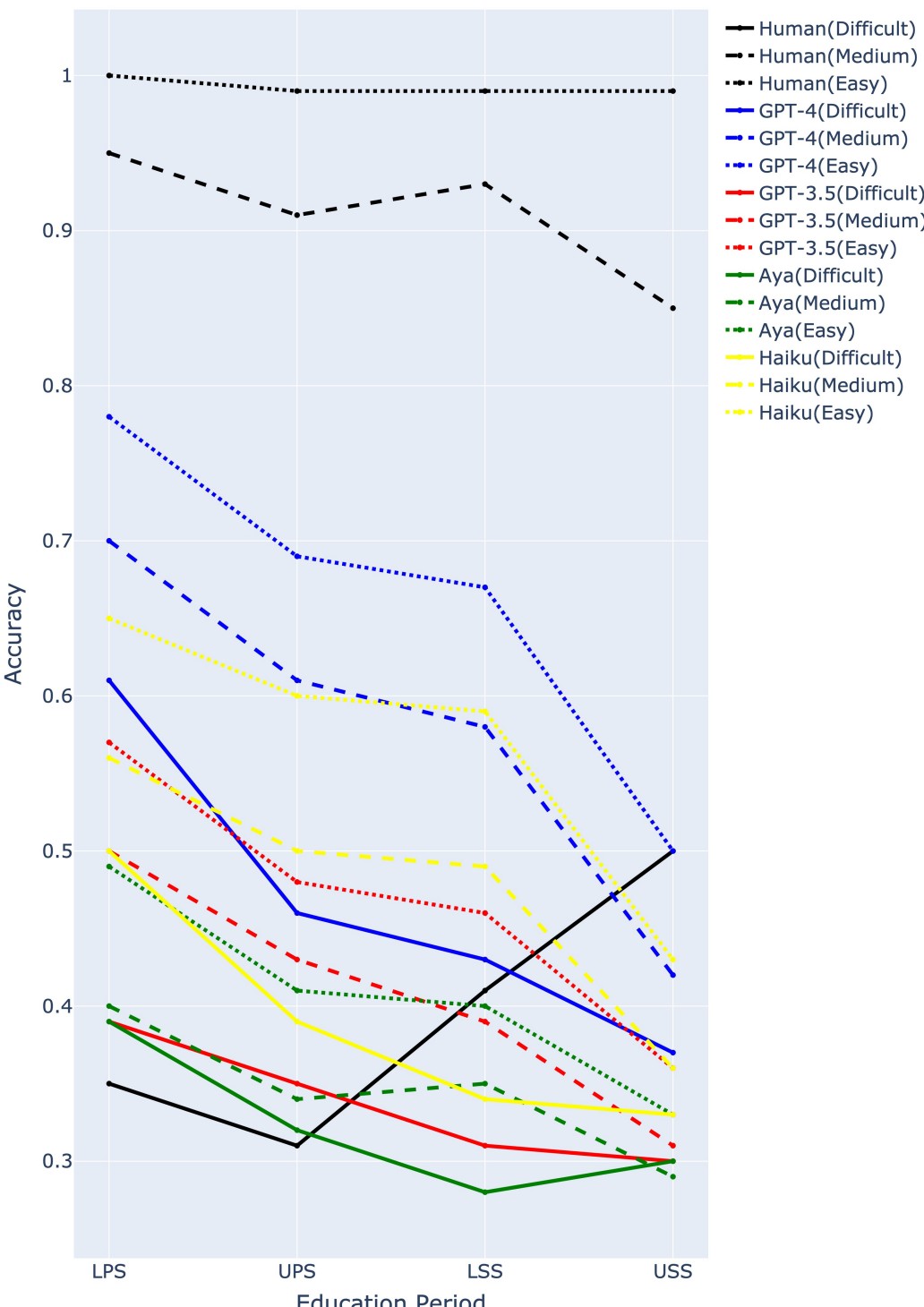

Figure 9: Accuracy of models and humans across three difficulty levels, segmented by question educational stage.

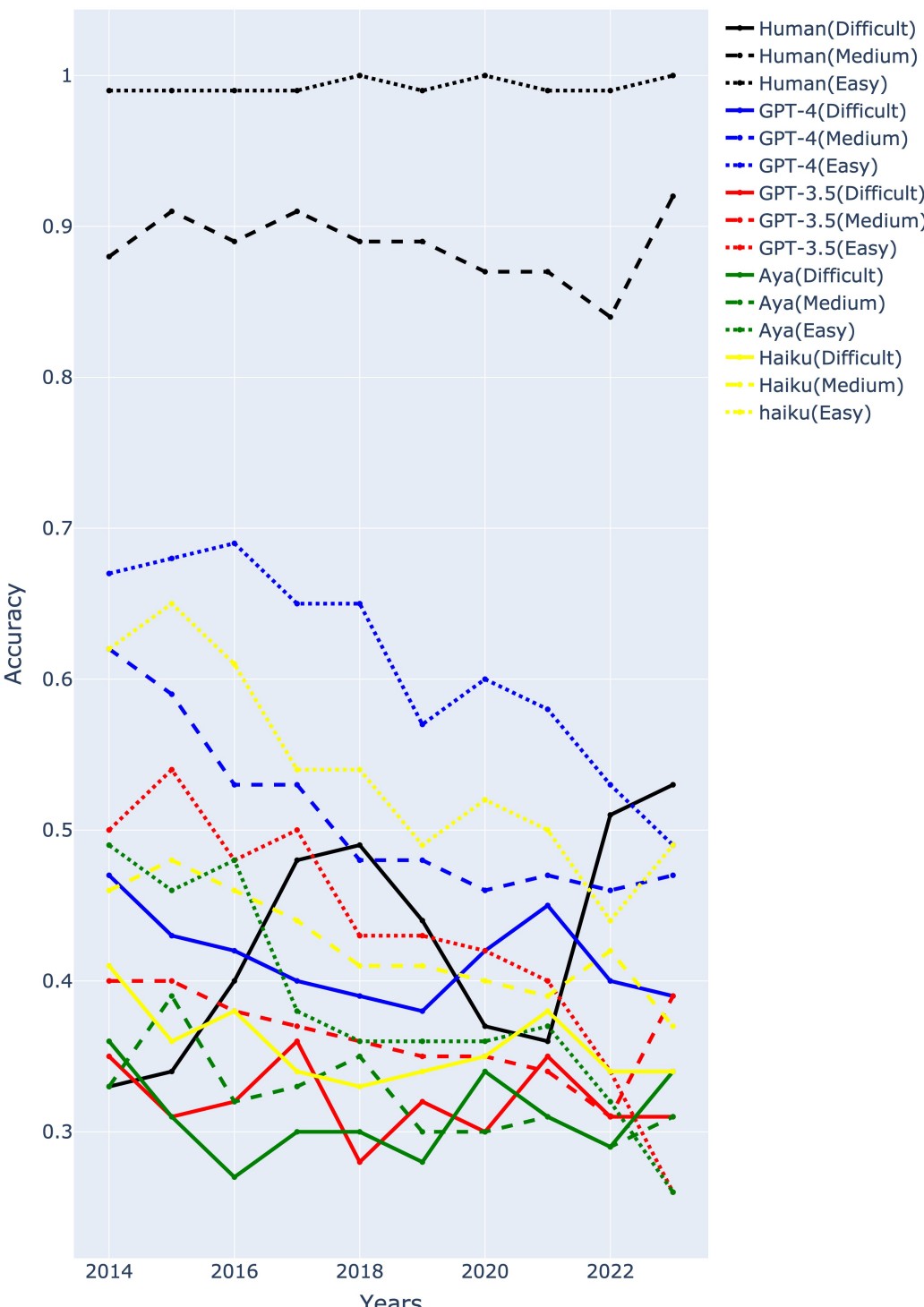

Figure 10: Accuracy of models and humans across three difficulty levels, segmented by question publication year.

| Main Category | Difficulty Level | Human | GPT-4 | Haiku | GPT-3.5 | Aya | XVERSE-13B | XVERSE-7B | PersianMind | mT0XL | mGPT | Random |
|---|---|---|---|---|---|---|---|---|---|---|---|---|
| Humanities | Difficult | 0.34 | 0.4 | 0.36 | 0.31 | 0.29 | 0.28 | 0.26 | 0.26 | 0.27 | 0.25 | 0.24 |
| Humanities | Easy | 0.99 | 0.61 | 0.54 | 0.39 | 0.36 | 0.32 | 0.3 | 0.28 | 0.3 | 0.22 | 0.21 |
| Humanities | Medium | 0.86 | 0.5 | 0.45 | 0.34 | 0.32 | 0.31 | 0.28 | 0.28 | 0.29 | 0.26 | 0.26 |
| Mathematics | Difficult | 0.36 | 0.36 | 0.31 | 0.3 | 0.27 | 0.26 | 0.28 | 0.25 | 0.26 | 0.25 | 0.24 |
| Mathematics | Easy | 0.99 | 0.49 | 0.38 | 0.35 | 0.3 | 0.29 | 0.3 | 0.28 | 0.26 | 0.26 | 0.24 |
| Mathematics | Medium | 0.84 | 0.41 | 0.33 | 0.32 | 0.27 | 0.27 | 0.27 | 0.26 | 0.26 | 0.25 | 0.25 |
| Natural Science | Difficult | 0.5 | 0.4 | 0.33 | 0.32 | 0.31 | 0.28 | 0.28 | 0.27 | 0.27 | 0.26 | 0.25 |
| Natural Science | Easy | 1 | 0.67 | 0.6 | 0.48 | 0.41 | 0.38 | 0.33 | 0.33 | 0.3 | 0.25 | 0.25 |
| Natural Science | Medium | 0.92 | 0.47 | 0.39 | 0.34 | 0.3 | 0.31 | 0.27 | 0.28 | 0.28 | 0.25 | 0.26 |
| Social Science | Difficult | 0.58 | 0.54 | 0.46 | 0.36 | 0.38 | 0.31 | 0.27 | 0.29 | 0.29 | 0.25 | 0.25 |
| Social Science | Easy | 1 | 0.74 | 0.68 | 0.55 | 0.5 | 0.41 | 0.4 | 0.37 | 0.38 | 0.23 | 0.27 |
| Social Science | Medium | 0.97 | 0.68 | 0.6 | 0.46 | 0.43 | 0.36 | 0.34 | 0.33 | 0.33 | 0.24 | 0.24 |
| Avg on all tasks | Avg on all tasks | 0.78 | 0.52 | 0.45 | 0.38 | 0.34 | 0.32 | 0.3 | 0.29 | 0.29 | 0.25 | 0.25 |
| Avg on all questions | Avg on all questions | 0.76 | 0.49 | 0.41 | 0.36 | 0.32 | 0.3 | 0.29 | 0.28 | 0.28 | 0.25 | 0.25 |

Table 13: Human vs model accuracy for all tasks: model answers extracted via "Regex" method

| Main Category | Diffuculty Level | Human | GPT-4 | Haiku | GPT-3.5 | Aya | XVERSE-13B | XVERSE-7B | PersianMind | mT0XL | mGPT | Random |
|---|---|---|---|---|---|---|---|---|---|---|---|---|
| Humanities | Difficult | 0.34 | 0.4 | 0.36 | 0.31 | 0.29 | 0.27 | 0.26 | 0.26 | 0.26 | 0.24 | 0.24 |
| Humanities | Easy | 0.99 | 0.61 | 0.54 | 0.39 | 0.37 | 0.31 | 0.3 | 0.28 | 0.29 | 0.24 | 0.21 |
| Humanities | Medium | 0.86 | 0.5 | 0.45 | 0.34 | 0.33 | 0.31 | 0.28 | 0.26 | 0.28 | 0.22 | 0.26 |
| Mathematics | Difficult | 0.36 | 0.36 | 0.31 | 0.3 | 0.26 | 0.25 | 0.26 | 0.25 | 0.27 | 0.23 | 0.24 |
| Mathematics | Easy | 0.99 | 0.49 | 0.38 | 0.35 | 0.29 | 0.27 | 0.29 | 0.26 | 0.27 | 0.23 | 0.24 |
| Mathematics | Medium | 0.84 | 0.41 | 0.33 | 0.32 | 0.27 | 0.26 | 0.27 | 0.26 | 0.27 | 0.23 | 0.25 |
| Natural Science | Difficult | 0.5 | 0.4 | 0.33 | 0.32 | 0.31 | 0.28 | 0.28 | 0.26 | 0.28 | 0.22 | 0.25 |
| Natural Science | Easy | 1 | 0.67 | 0.6 | 0.48 | 0.41 | 0.37 | 0.31 | 0.29 | 0.28 | 0.19 | 0.25 |
| Natural Science | Medium | 0.92 | 0.47 | 0.39 | 0.34 | 0.31 | 0.3 | 0.28 | 0.25 | 0.29 | 0.21 | 0.26 |
| Social Science | Difficult | 0.58 | 0.54 | 0.46 | 0.36 | 0.37 | 0.3 | 0.27 | 0.26 | 0.3 | 0.21 | 0.25 |
| Social Science | Easy | 1 | 0.74 | 0.68 | 0.55 | 0.5 | 0.41 | 0.4 | 0.34 | 0.33 | 0.22 | 0.27 |
| Social Science | Medium | 0.97 | 0.68 | 0.6 | 0.46 | 0.42 | 0.35 | 0.34 | 0.32 | 0.32 | 0.23 | 0.24 |
| Avg on all tasks | Avg on all tasks | 0.78 | 0.52 | 0.45 | 0.38 | 0.34 | 0.31 | 0.29 | 0.27 | 0.29 | 0.22 | 0.25 |
| Avg on all questions | Avg on all questions | 0.76 | 0.49 | 0.41 | 0.36 | 0.32 | 0.29 | 0.28 | 0.27 | 0.28 | 0.22 | 0.25 |

Table 14: Human vs model accuracy for all tasks: model answers extracted via "Single Token Probability" method

| Main Category | Difficulty Level | Human | GPT-4 | Haiku | GPT-3.5 | Aya | XVERSE-13B | XVERSE-7B | PersianMind | mT0XL | mGPT | Random |
|---|---|---|---|---|---|---|---|---|---|---|---|---|
| Humanities | Difficult | 0.34 | 0.4 | 0.36 | 0.31 | 0.25 | 0.28 | 0.26 | 0.26 | 0.25 | 0.24 | 0.24 |
| Humanities | Easy | 0.99 | 0.61 | 0.54 | 0.39 | 0.3 | 0.32 | 0.29 | 0.25 | 0.28 | 0.23 | 0.21 |
| Humanities | Medium | 0.86 | 0.5 | 0.45 | 0.34 | 0.26 | 0.31 | 0.28 | 0.26 | 0.26 | 0.24 | 0.26 |
| Mathematics | Difficult | 0.36 | 0.36 | 0.31 | 0.3 | 0.23 | 0.26 | 0.26 | 0.25 | 0.24 | 0.23 | 0.24 |
| Mathematics | Easy | 0.99 | 0.49 | 0.38 | 0.35 | 0.25 | 0.26 | 0.28 | 0.24 | 0.25 | 0.23 | 0.24 |
| Mathematics | Medium | 0.84 | 0.41 | 0.33 | 0.32 | 0.25 | 0.25 | 0.26 | 0.25 | 0.23 | 0.24 | 0.25 |
| Natural Science | Difficult | 0.5 | 0.4 | 0.33 | 0.32 | 0.27 | 0.28 | 0.28 | 0.27 | 0.26 | 0.24 | 0.25 |
| Natural Science | Easy | 1 | 0.67 | 0.6 | 0.48 | 0.32 | 0.37 | 0.32 | 0.25 | 0.27 | 0.23 | 0.25 |
| Natural Science | Medium | 0.92 | 0.47 | 0.39 | 0.34 | 0.29 | 0.3 | 0.28 | 0.24 | 0.26 | 0.25 | 0.26 |
| Social Science | Difficult | 0.58 | 0.54 | 0.46 | 0.36 | 0.3 | 0.3 | 0.27 | 0.27 | 0.27 | 0.25 | 0.25 |
| Social Science | Easy | 1 | 0.74 | 0.68 | 0.55 | 0.4 | 0.42 | 0.4 | 0.28 | 0.32 | 0.24 | 0.27 |
| Social Science | Medium | 0.97 | 0.68 | 0.6 | 0.46 | 0.34 | 0.36 | 0.34 | 0.29 | 0.29 | 0.24 | 0.24 |
| Avg on all tasks | Avg on all tasks | 0.78 | 0.52 | 0.45 | 0.38 | 0.29 | 0.31 | 0.29 | 0.26 | 0.27 | 0.24 | 0.25 |
| Avg on all questions | Avg on all questions | 0.76 | 0.49 | 0.41 | 0.36 | 0.28 | 0.29 | 0.28 | 0.26 | 0.26 | 0.24 | 0.25 |

Table 15: Human vs model accuracy for all tasks: model answers extracted via "Full Answer Probability" method

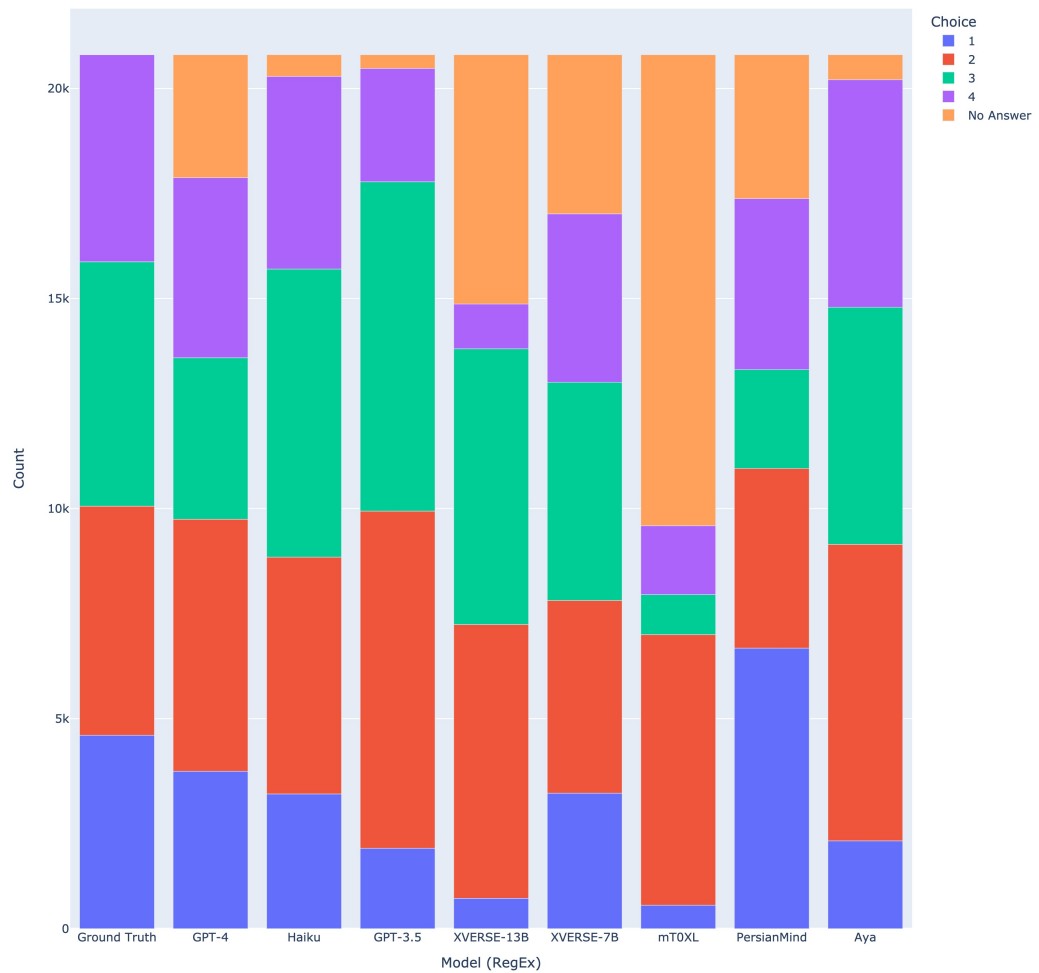

Figure 11: Selected choice distribution of different models: model answers extracted via "Regex" method

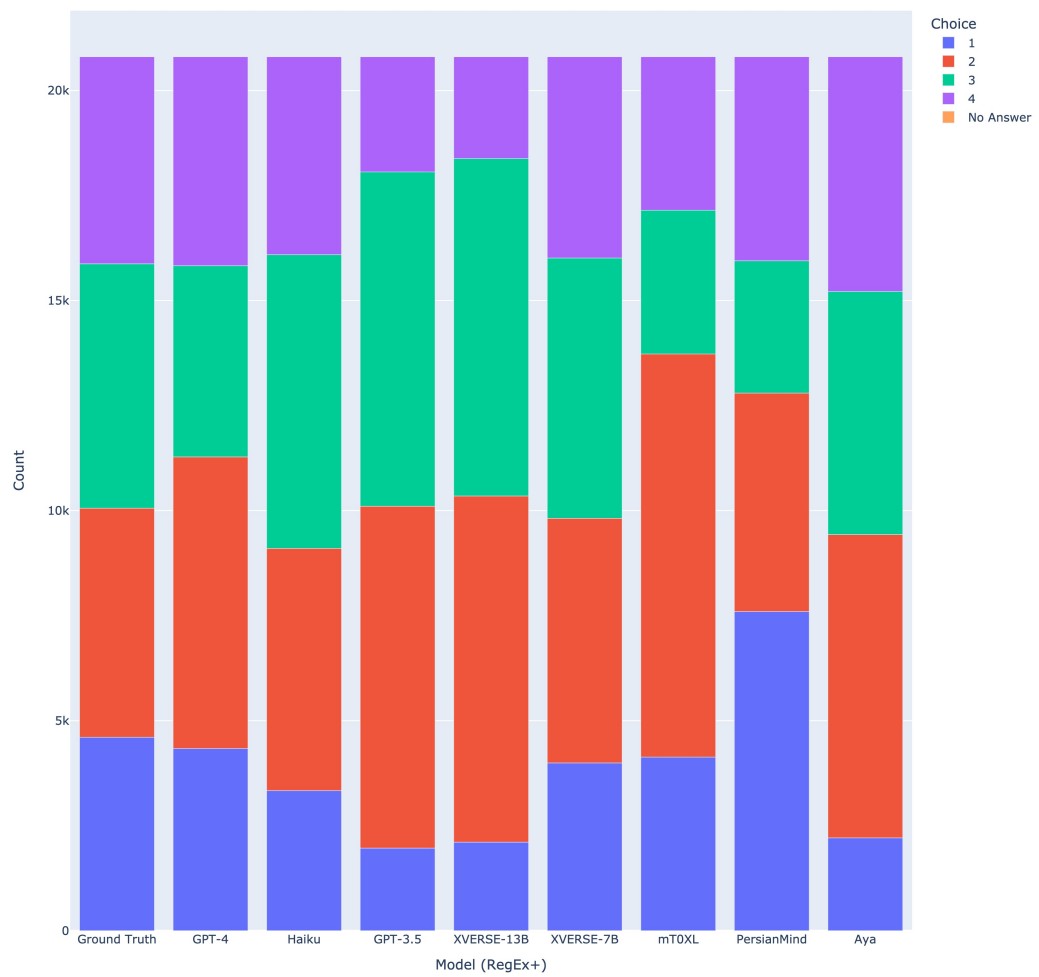

Figure 12: Selected choice distribution of different models: model answers extracted via "Regex" method

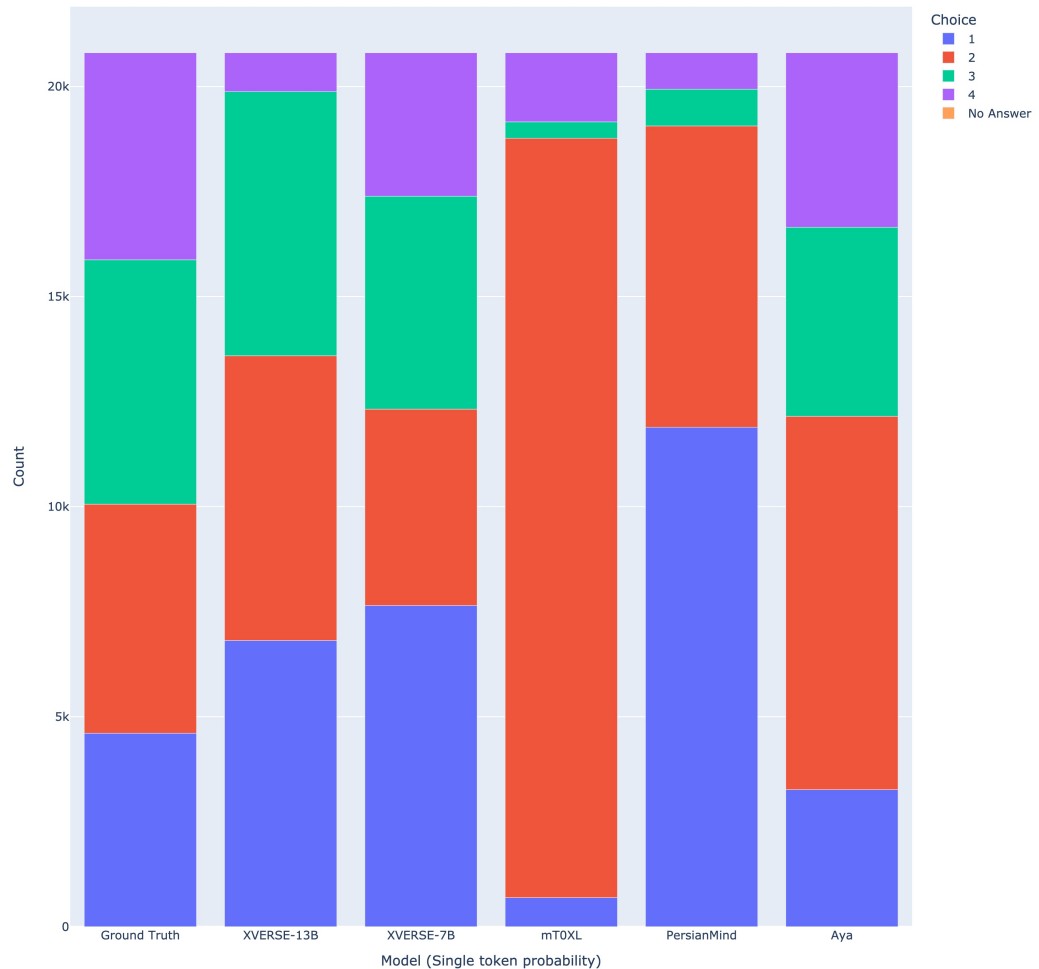

Figure 13: Selected choice distribution of different models: model answers extracted via "Single Token Probability" method

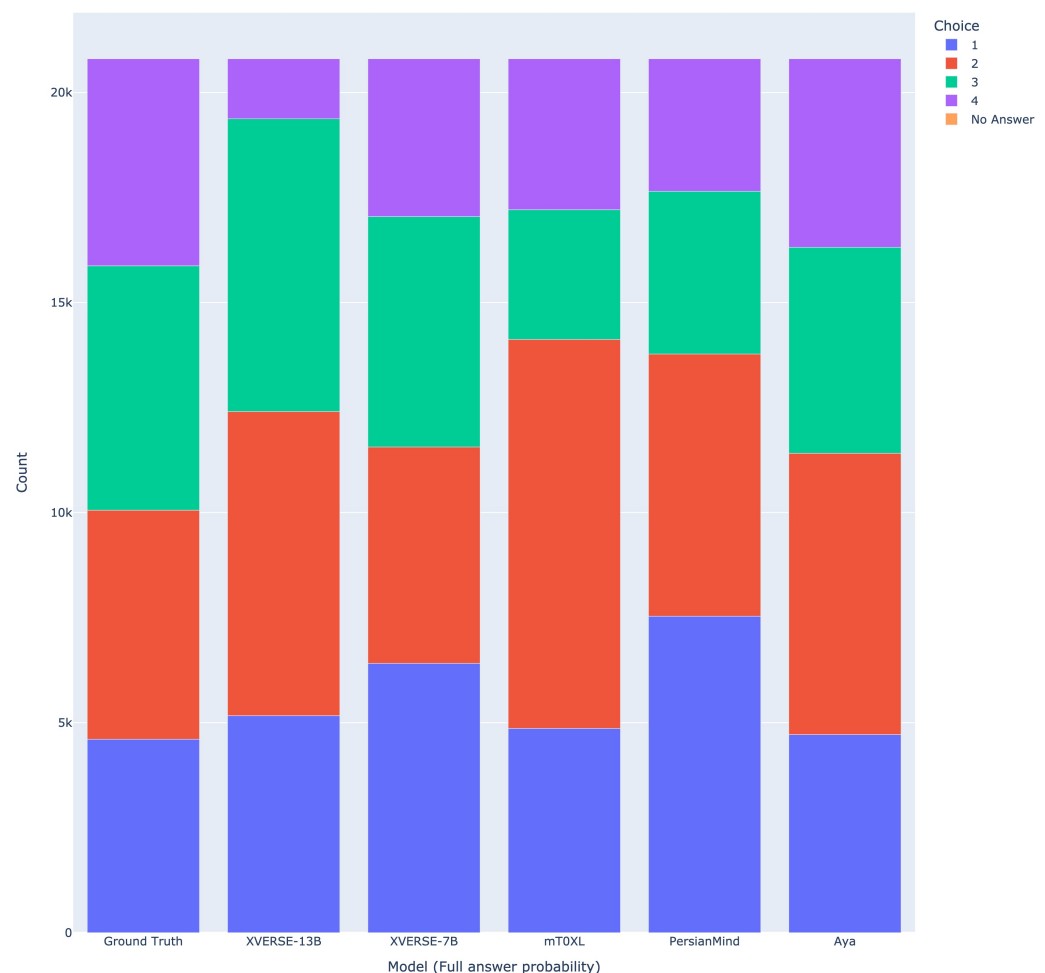

Figure 14: Selected choice distribution of different models: model answers extracted via "Full Answer Probability" method

# F   Trap

| Main Category | Human | GPT-4 | Haiku | GPT-3.5 | Aya | XVERSE-13B | XVERSE-7B | PersianMind | mT0XL | mGPT | Random |
|---|---|---|---|---|---|---|---|---|---|---|---|
| Humanities | 0.71 / 0.22 | 0.47 / 0.32 | 0.43 / 0.31 | 0.33 / 0.29 | 0.31 / 0.29 | 0.3 / 0.27 | 0.27 / 0.23 | 0.28 / 0.25 | 0.28 / 0.25 | 0.25 / 0.24 | 0.25 / 0.27 |
| Mathematics | 0.72 / 0.23 | 0.41 / 0.4 | 0.33 / 0.29 | 0.32 / 0.31 | 0.28 / 0.27 | 0.27 / 0.26 | 0.28 / 0.29 | 0.26 / 0.24 | 0.26 / 0.26 | 0.25 / 0.26 | 0.25 / 0.25 |
| Natural Science | 0.79 / 0.31 | 0.49 / 0.42 | 0.42 / 0.31 | 0.37 / 0.29 | 0.33 / 0.3 | 0.32 / 0.26 | 0.29 / 0.26 | 0.29 / 0.26 | 0.28 / 0.27 | 0.25 / 0.26 | 0.25 / 0.26 |
| Social Science | 0.85 / 0.35 | 0.64 / 0.42 | 0.57 / 0.38 | 0.44 / 0.29 | 0.42 / 0.35 | 0.35 / 0.28 | 0.33 / 0.27 | 0.32 / 0.26 | 0.33 / 0.27 | 0.24 / 0.29 | 0.25 / 0.22 |
| Avg on all tasks | 0.77 / 0.28 | 0.5 / 0.39 | 0.44 / 0.32 | 0.37 / 0.3 | 0.34 / 0.3 | 0.31 / 0.27 | 0.29 / 0.26 | 0.29 / 0.25 | 0.29 / 0.26 | 0.25 / 0.26 | 0.25 / 0.25 |
| Avg on all questions | 0.76 / 0.26 | 0.49 / 0.39 | 0.41 / 0.31 | 0.36 / 0.3 | 0.32 / 0.29 | 0.3 / 0.27 | 0.29 / 0.26 | 0.28 / 0.25 | 0.28 / 0.26 | 0.25 / 0.26 | 0.25 / 0.25 |

Table 16: Accuracy of humans and various models on main categories, including all questions/trapped questions.

# G   Statistical tests

| Model | T-Test_response_str | T-Test_Question Body | T-Test |
|---|---|---|---|
| XVERSE-13B | -4.52 / 0.0 | -3.18 / 0.0 | -7.52 / 0.0 |
| mGPT | -0.13 / 0.9 | 0.06 / 0.95 | 2.35 / 0.02 |
| PersianMind | -0.37 / 0.71 | -2.4 / 0.02 | -3.65 / 0.0 |
| GPT-4 | 7.2 / 0.0 | -11.54 / 0.0 | -26.16 / 0.0 |
| Aya | 1.28 / 0.2 | -4.36 / 0.0 | -11.03 / 0.0 |
| Haiku | -11.28 / 0.0 | -13.08 / 0.0 | -22.56 / 0.0 |
| mT0XL | -0.3 / 0.77 | -0.65 / 0.52 | -4.67 / 0.0 |
| XVERSE-7B | -0.19 / 0.85 | -3.05 / 0.0 | -6.51 / 0.0 |
| GPT-3.5 | -1.44 / 0.15 | -7.03 / 0.0 | -14.22 / 0.0 |
| Human | NA | -11.87 / 2.17 | -12.92 / 5.08 |

Table 17: T-test comparison of question and response string lengths among the models

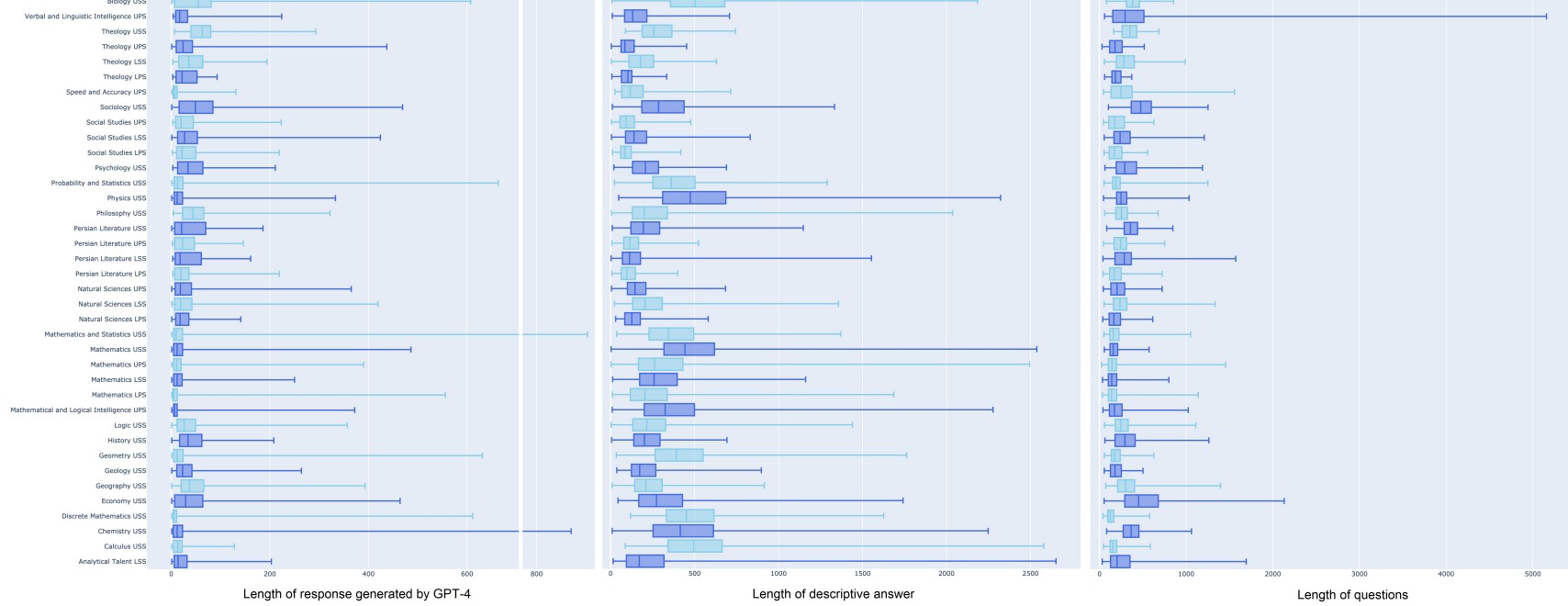

Figure 15: Comparing the lengths of GPT-4 generated strings, descriptive answers, and questions

## H   Sample questions

توابع $f$ و $g$ با دامنه اعداد طبیعی به صورت $f(x) = (-1)^x \left[\frac{x+1}{2}\right]$ و $g(x) = x - (-1)^x$ تعریف شده‌اند. حاصل $(go(-f))^{-1}(۱۰۱) - (gof)^{-1}(۱۰۲)$ کدام است؟ ($[\ ]$، نماد جزء صحیح است.)

(۱). صفر

(۲). ۱

(۳). ۲

(۴). ۳

The functions $f$ and $g$ are defined on the domain of natural numbers as follows: $f(x) = (-1)^x \left[\frac{x+1}{2}\right]$ and $g,(x) = x, -(-1)^x$. What is the result of $(gof)^{-1}(101) - (go,(-f))^{-1}(102)$? ([] denotes the floor function).

1. zero
2. 1
3. 2
4. 3

**Mathematics**  **USS**  **Trap:-**  **Difficult**

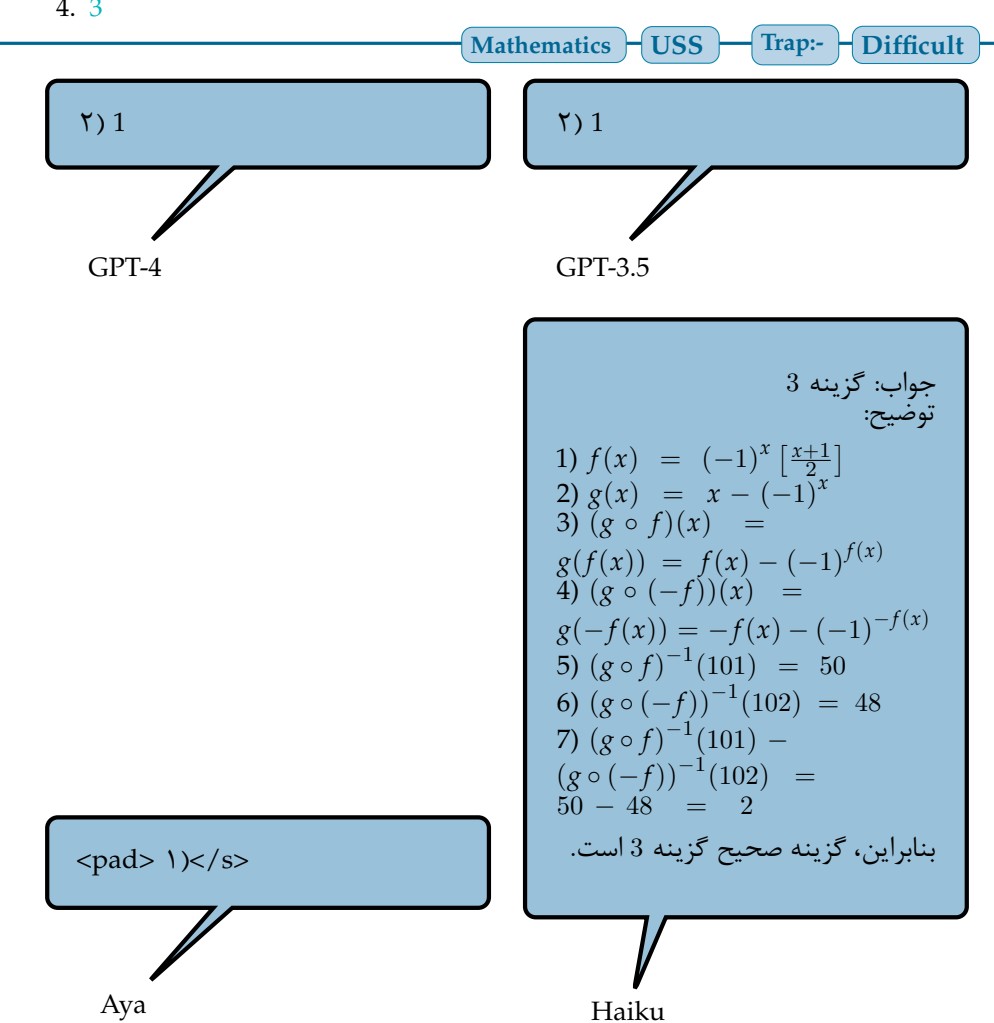

۲) 1

GPT-4

۲) 1

GPT-3.5

جواب: گزینه 3
توضیح:

1) $f(x) = (-1)^x \left[\frac{x+1}{2}\right]$
2) $g(x) = x - (-1)^x$
3) $(g \circ f)(x) = g(f(x)) = f(x) - (-1)^{f(x)}$
4) $(g \circ (-f))(x) = g(-f(x)) = -f(x) - (-1)^{-f(x)}$
5) $(g \circ f)^{-1}(101) = 50$
6) $(g \circ (-f))^{-1}(102) = 48$
7) $(g \circ f)^{-1}(101) - (g \circ (-f))^{-1}(102) = 50 - 48 = 2$

بنابراین، گزینه صحیح گزینه 3 است.

<pad> ۱)

Aya

Haiku

عبارت «ما تمایل داریم اشیای ناقص را بهصورت کامل ببینیم»، به کدام اصل از اصول گشتالت اشاره
می‌کند؟
(۱). مشابهت
(۲). تکمیل
(۳). استمرار
(۴). مجاورت

The phrase "We tend to perceive incomplete objects as complete" refers to which
principle of Gestalt principles?

1. Similarity
2. Closure
3. Continuity
4. Proximity

Psychology — USS — Trap: - — Easy

۲) تکمیل

۲) تکمیل

GPT-4

GPT-3.5

پاسخ صحیح گزینه 2) تکمیل است.
اصل تکمیل در روانشناسی گشتالت
به این اشاره دارد که ما تمایل داریم
اشیای ناقص را به صورت کامل
ببینیم و کمبودها و شکاف‌های
موجود را ذهناً پر کنیم. این اصل
توضیح می‌دهد چرا ما اغلب اشیای
ناقص را به صورت کامل درک
می‌کنیم.
سایر گزینه‌ها به اصول دیگر گشتالت
اشاره دارند:
1) مشابهت: اشیای مشابه را به صورت
گروه‌های واحد درک می‌کنیم.
3) استمرار: ما تمایل داریم خطوط و
اشکال را به صورت یکپارچه و پیوسته
ببینیم.
4) مجاورت: اشیای نزدیک به هم را
به صورت گروه‌های واحد درک
می‌کنیم.

<pad> ۲)

Aya

Haiku

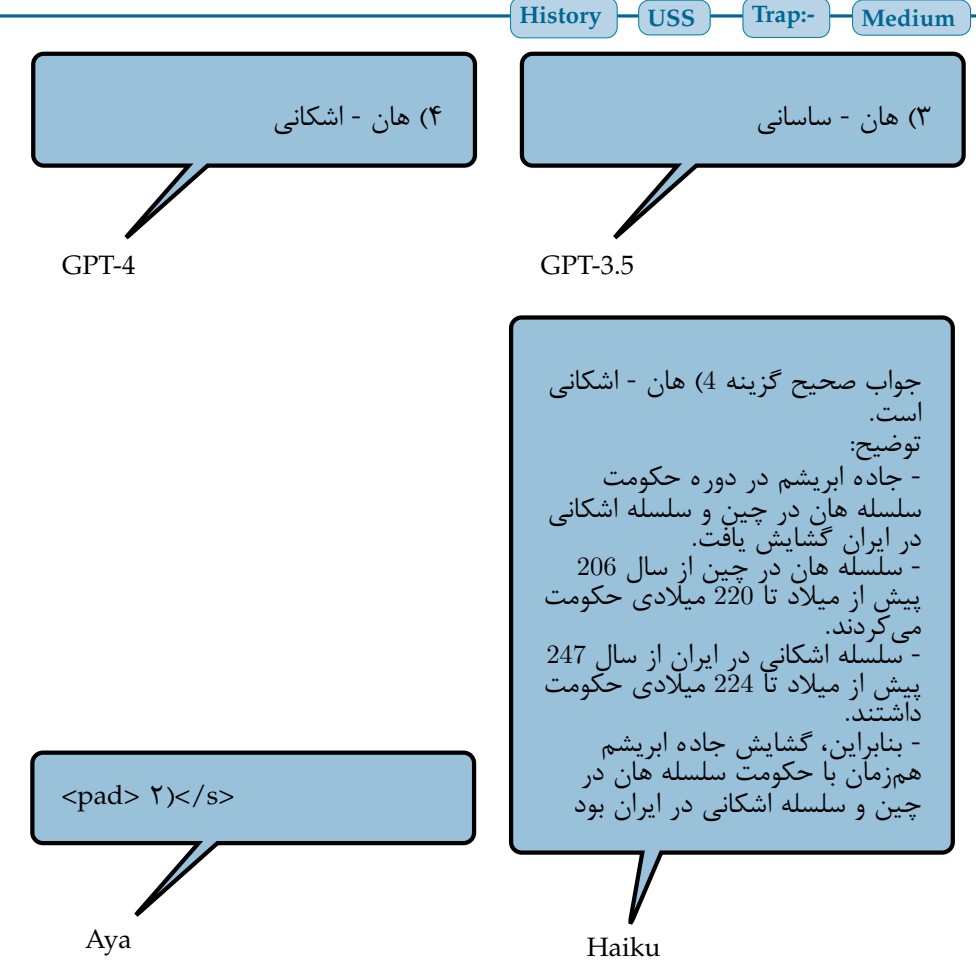

یک کالا طی سه مرحله تولید و در نهایت به مشتری فروخته می‌شود. اگر در مرحله آخر (سوم) قیمت کالا برابر با ۱۲,۵۰۰ تومان باشد و اعتبار مالیات بر ارزش افزوده‌هایی که در مرحله قبل پرداخت شده برابر با ۵۵۰ تومان باشد، آنگاه:

(الف)   قیمت فروش کالا با مالیات بر ارزش افزوده در مرحله سوم چند تومان است ؟

(ب)   قیمت فروش کالا با مالیات بر ارزش افزوده در مرحله دوم چند تومان است؟

(ج)   مجموع مالیات بر ارزش افزوده جمع‌آوری شده چند تومان است؟ (نرخ مالیات بر ارزش افزوده ۵ درصد است.)

(۱). الف) ۱۳,۱۲۵، ب) ۱۰,۵۰۰، ج) ۸۷۵

(۲). الف) ۱۳,۹۸۱، ب) ۱۱,۵۵۰، ج) ۸۷۵

(۳). الف) ۱۳,۱۲۵، ب) ۱۱,۵۵۰، ج) ۶۲۵

(۴). الف) ۱۳,۹۸۱، ب) ۱۰,۵۰۰، ج) ۶۲۵

A product goes through three stages of production and is ultimately sold to the customer. If the price of the product in the final (third) stage is 12, 500 Tomans and the credit for the value-added taxes paid in the previous stage is 550 Tomans, then:

(a) What is the selling price of the product with value-added tax in the third stage?

(b) What is the selling price of the product with value-added tax in the second stage?

(c) What is the total value-added tax collected?

1. (a)13, 125, (b)10, 500, (c)875

2. (a)13, 981, (b)11, 550, (c)875

3. (a)13, 125, (b)11, 550, (c)625

4. (a)13, 981, (b)10, 500, (c)625

Economy — USS — Trap:- — Difficult

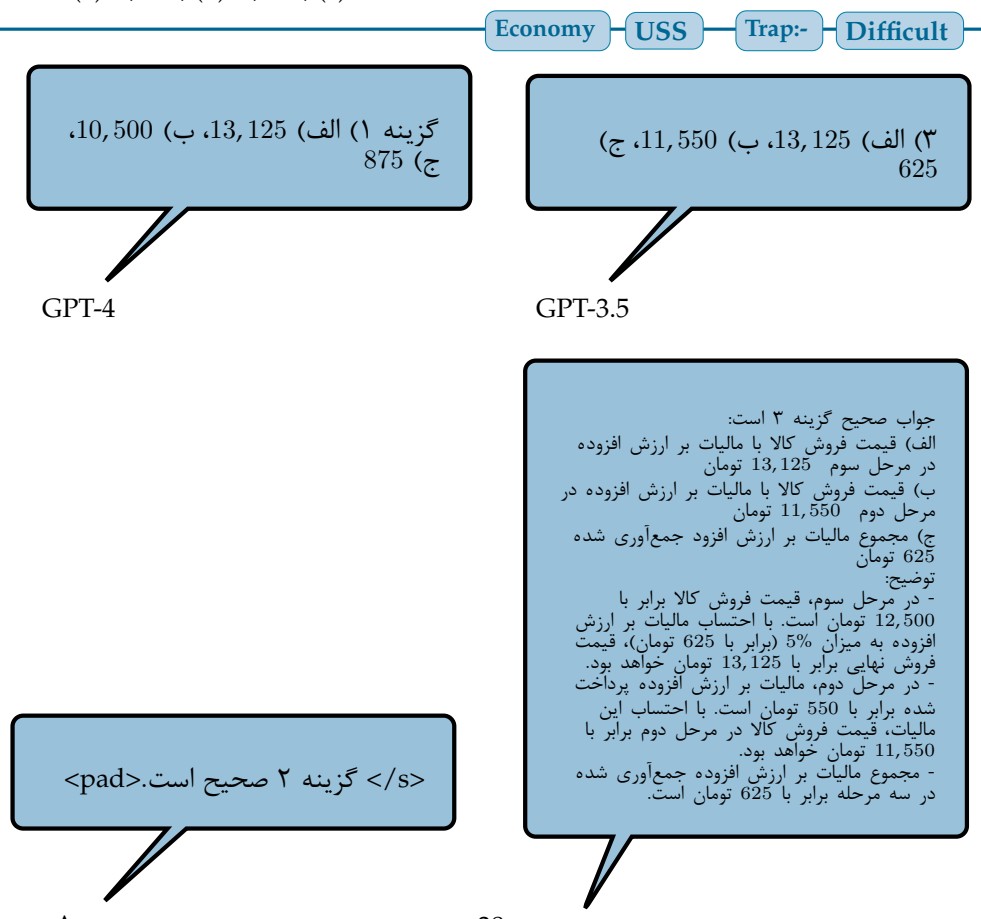

گزینه ۱) الف) 13,125، ب) 10,500، ج) 875

GPT-4

۳) الف) 13,125، ب) 11,550، ج) 625

GPT-3.5

جواب صحیح گزینه ۳ است:
الف) قیمت فروش کالا با مالیات بر ارزش افزوده در مرحل سوم   13,125 تومان
ب) قیمت فروش کالا با مالیات بر ارزش افزوده در مرحل دوم   11,550 تومان
ج) مجموع مالیات بر ارزش افزود جمع‌آوری شده   625 تومان
توضیح:
- در مرحل سوم، قیمت فروش کالا برابر با 12,500 تومان است. با احتساب مالیات بر ارزش افزوده به میزان 5% (برابر با 625 تومان)، قیمت فروش نهایی برابر با 13,125 تومان خواهد بود.
- در مرحل دوم، مالیات بر ارزش افزوده پرداخت شده برابر با 550 تومان است. با احتساب این مالیات، قیمت فروش کالا در مرحل دوم برابر با 11,550 تومان خواهد بود.
- مجموع مالیات بر ارزش افزوده جمع‌آوری شده در سه مرحله برابر با 625 تومان است.

گزینه ۲ صحیح است.<pad>

Aya

   Haiku

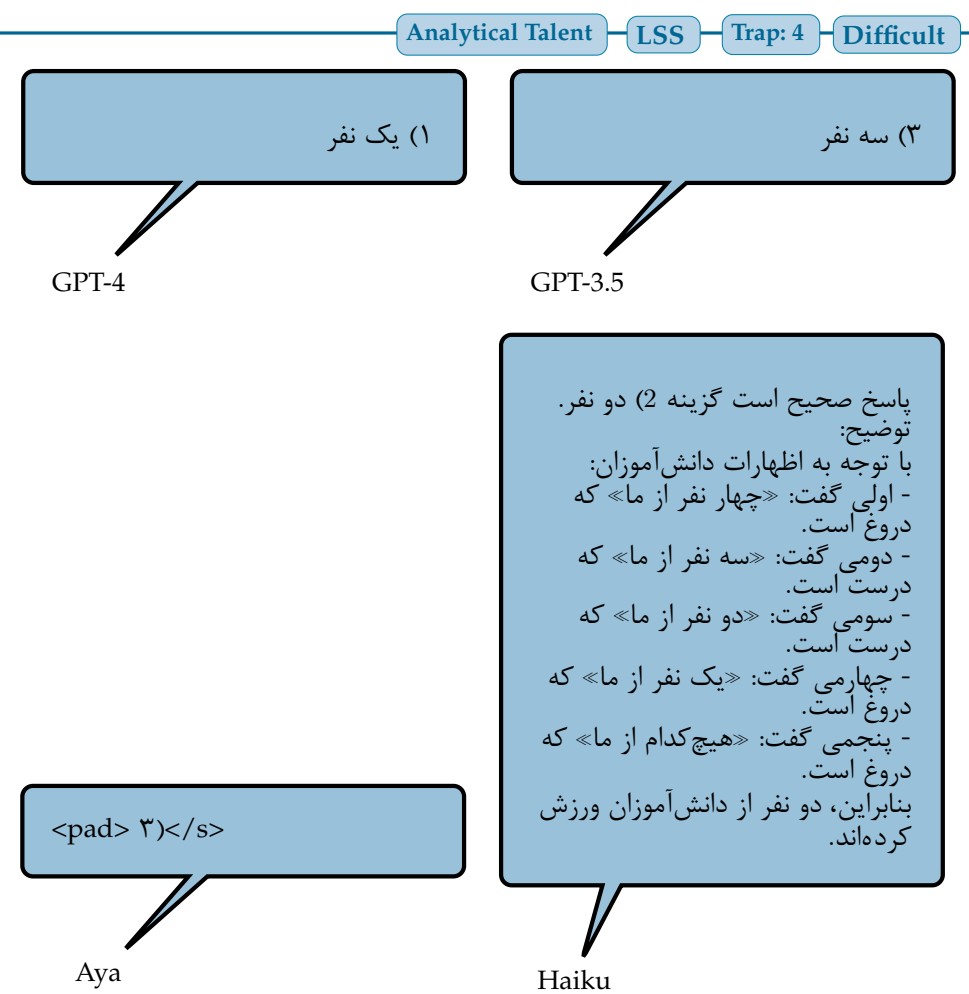

معلمی وارد کلاس پنج‌نفره شد و از دانش‌آموزان پرسید چند نفر دیروز ورزش کردند. اولی گفت: «چهار نفر از ما»، دومی گفت: «سه نفر از ما»، سومی گفت: «دو نفر از ما»، چهارمی گفت: «یک نفر از ما» و پنجمی گفت: «هیچ‌کدام از ما». معلم می‌دانست کسانی که ورزش کرده‌اند راست و کسانی که ورزش نکرده‌اند، دروغ می‌گویند. چند نفر ورزش کرده‌اند؟

(۱). یک نفر

(۲). دو نفر

(۳). سه نفر

(۴). کسی ورزش نکرده است.

A teacher entered a class of five students and asked how many of them had exercised yesterday. The first student said, "Four of us," the second said, "Three of us," the third said, "Two of us," the fourth said, "One of us," and the fifth said, "None of us." The teacher knew that those who had exercised were telling the truth, while those who hadn't were lying. How many students had exercised?

1. One student
2. Two students
3. Three students
4. No student has exercised

Analytical Talent · LSS · Trap: 4 · Difficult

۱) یک نفر

GPT-4

۳) سه نفر

GPT-3.5

پاسخ صحیح است گزینه 2) دو نفر.
توضیح:
با توجه به اظهارات دانش‌آموزان:
- اولی گفت: «چهار نفر از ما» که دروغ است.
- دومی گفت: «سه نفر از ما» که درست است.
- سومی گفت: «دو نفر از ما» که درست است.
- چهارمی گفت: «یک نفر از ما» که دروغ است.
- پنجمی گفت: «هیچ‌کدام از ما» که دروغ است.
بنابراین، دو نفر از دانش‌آموزان ورزش کرده‌اند.

<pad> ۳)

Aya

Haiku

عبارات زیر به‌ترتیب مصادیق کدام موارد هستند؟
- کنش ناظر به دیگران
- گوش دادن آهنگ در اتوبوس بدون توجه به سایر مسافران
- احترام به سالمندان

(۱). کنش اجتماعی - کنش فردی - ارزش اجتماعی
(۲). کنش اجتماعی - کنش فردی - هنجار اجتماعی
(۳). کنش فردی - کنش درونی - هنجار اجتماعی
(۴). کنش فردی - کنش درونی - ارزش اجتماعی

The following phrases correspond to which of the following scenarios?

- Observing others' actions
- Listening to music on the bus without regard to other passengers
- Respecting the elderly

1. Social action - Individual action - Social value
2. Social action - Individual action - Social norm
3. Individual action - Intrinsic action - Social norm
4. Individual action - Intrinsic action - Social value

**Sociology** **USS** **Trap: -** **Easy**

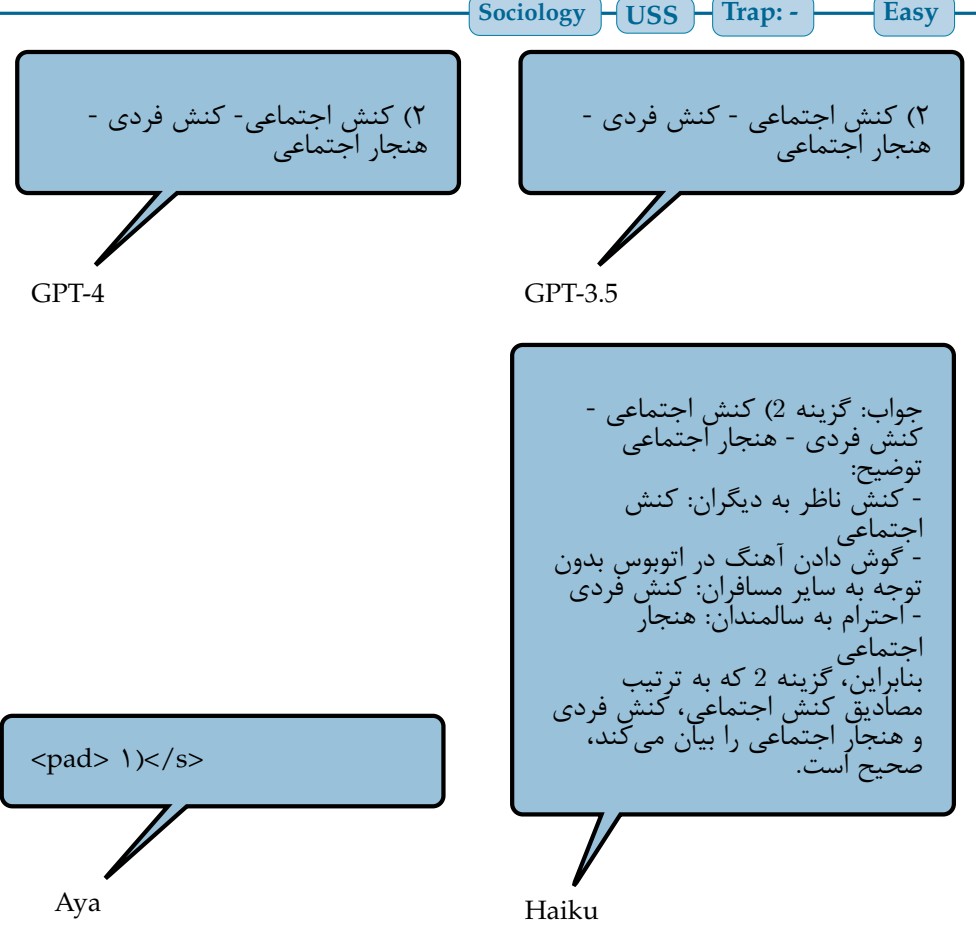

۲) کنش اجتماعی- کنش فردی - هنجار اجتماعی

GPT-4

۲) کنش اجتماعی - کنش فردی - هنجار اجتماعی

GPT-3.5

جواب: گزینه 2) کنش اجتماعی - کنش فردی - هنجار اجتماعی
توضیح:
- کنش ناظر به دیگران: کنش اجتماعی
- گوش دادن آهنگ در اتوبوس بدون توجه به سایر مسافران: کنش فردی
- احترام به سالمندان: هنجار اجتماعی
بنابراین، گزینه 2 که به ترتیب مصادیق کنش اجتماعی، کنش فردی و هنجار اجتماعی را بیان می‌کند، صحیح است.

Haiku

<pad> ۱)

Aya

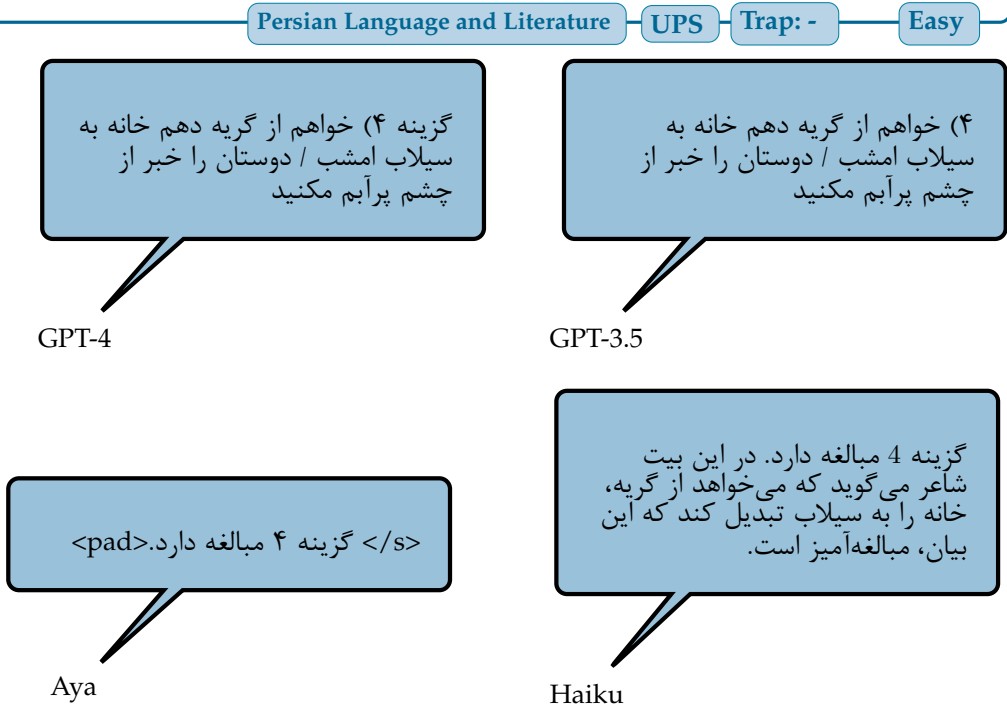

کدام بیت مبالغه دارد؟

(۱). حافظا چون غم و شادی جهان در گذر است / بهتر آن است که من خاطر خود خوش دارم

(۲). در کوی نیک‌نامی ما را گذر ندادند / گر تو نمی‌پسندی تغییر کن قضا را

(۳). روز وصل دوستداران یاد باد / یاد باد آن روزگاران یاد باد

(۴). خواهم از گریه دهم خانه به سیلاب امشب / دوستان را خبر از چشم پرآبم مکنید

Which verse has Hyperbole?

1. O Hafiz, since the world is passing through sorrow and joy, / It is better that I keep my heart content.
2. They did not let us pass through the alley of good reputation / If you don't like it, change your destiny.
3. May the day of reunion with beloved friends be remembered / Remember the days of old.
4. Tonight, I cry so much that it seems as if a flood has come into the house / Do not tell my friends that my eyes are full of tears.

Persian Language and Literature — UPS — Trap: - — Easy

**GPT-4**

گزینه ۴) خواهم از گریه دهم خانه به سیلاب امشب / دوستان را خبر از چشم پرآبم مکنید

**GPT-3.5**

۴) خواهم از گریه دهم خانه به سیلاب امشب / دوستان را خبر از چشم پرآبم مکنید

**Aya**

 گزینه ۴ مبالغه دارد.<pad>

**Haiku**

گزینه 4 مبالغه دارد. در این بیت شاعر می‌گوید که می‌خواهد از گریه، خانه را به سیلاب تبدیل کند که این بیان، مبالغه‌آمیز است.

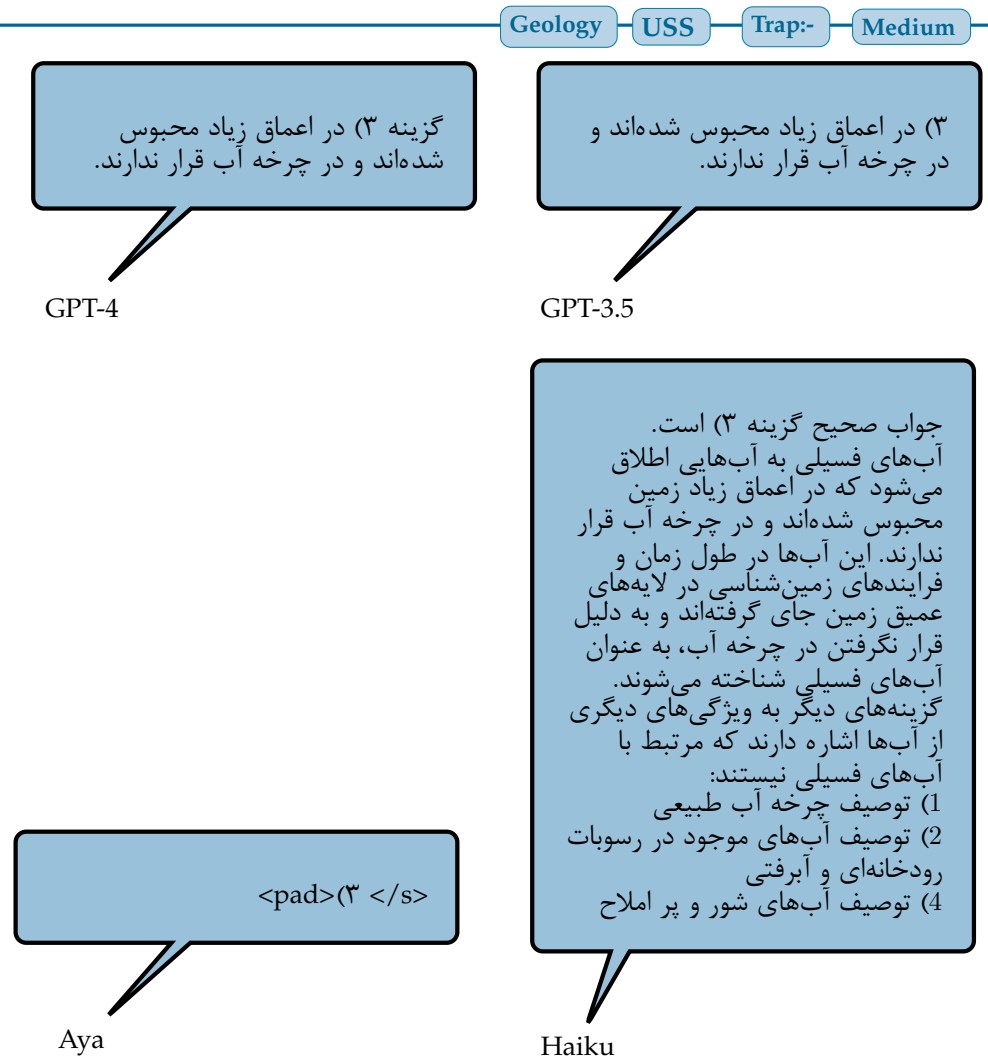

چند مورد، درباره یاخته‌های شرکت‌کننده در انعکاس عقب کشیدن دست فرد در برخورد با جسم داغ، درست است؟

- هر یاخته عصبی که پیام گیرنده درد را منتقل می‌کند، به بخش حرکتی دستگاه عصبی محیطی اختصاص دارد.

- بخشی از یاخته‌های عصبی که به عصب نخاعی تعلق دارند، با یاخته‌های چندهسته‌ای، ارتباط ویژه‌ای برقرار می‌کنند.

- هر یاخته عصبی که با عضله ناحیه بازو همایه (سیناپس) برقرار می‌کند، تغییری در پتانسیل الکتریکی آن رخ داده است.

- بعضی از یاخته‌های عصبی که جسم یاخته‌ای آنها در ماده خاکستری قرار دارد، با یاخته‌های عصبی حسی همایه (سیناپس) برقرار می‌کنند.

(۱). یک

(۲). دو

(۳). سه

(۴). چهار

How many statements about the neurons involved in the withdrawal reflex of the hand when encountering a hot object are correct?

- Each sensory neuron that transmits pain signals has a specific connection to the motor part of the peripheral nervous system.
- Some of the nerve fibers belonging to the spinal nerve establish a special connection with the multiple nerve fibers.
- Any sensory neuron that synapses with the muscle of the forearm changes its electrical potential.
- Some of the nerve fibers whose cell bodies are located in the dorsal root ganglion synapse with sensory nerve fibers.

1. One
2. Two
3. Three
4. Four

Biology — USS — Trap: - — Difficult

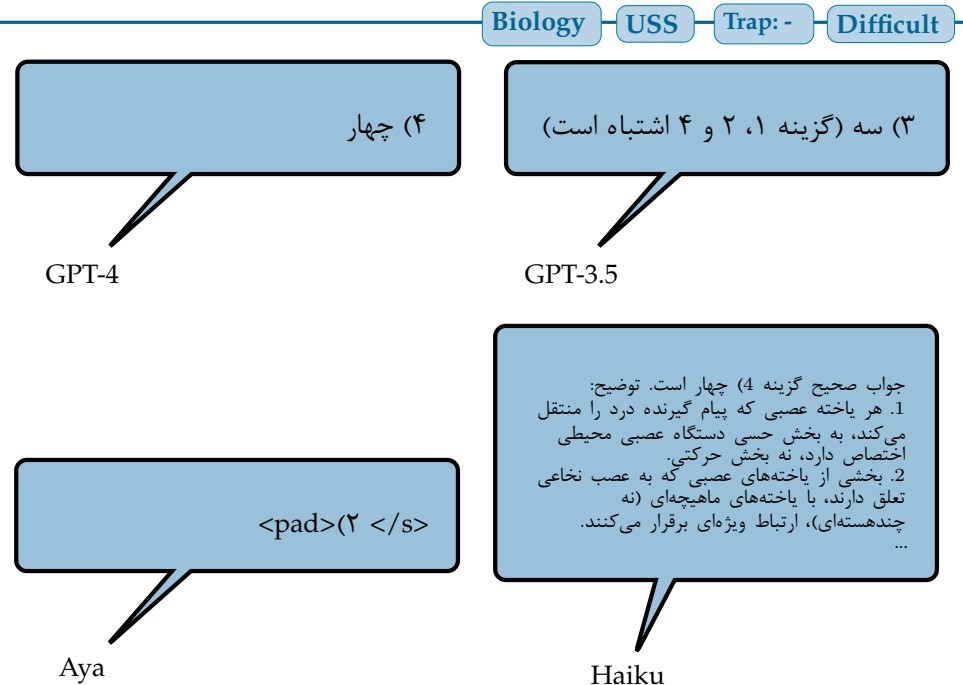

GPT-4: ۴) چهار

GPT-3.5: ۳) سه (گزینه ۱، ۲ و ۴ اشتباه است)

Aya: <pad>(۲ 

Haiku: جواب صحیح گزینه 4) چهار است. توضیح:
۱. هر یاخته عصبی که پیام گیرنده درد را منتقل می‌کند، به بخش حسی دستگاه عصبی محیطی اختصاص دارد، نه بخش حرکتی.
۲. بخشی از یاخته‌های عصبی که به عصب نخاعی تعلق دارند، با یاخته‌های ماهیچه‌ای (نه چندهسته‌ای)، ارتباط ویژه‌ای برقرار می‌کنند.
...

در عدد ۱۹۵۴۸۳، رقم یکان با یکان هزار، دهگان با دهگان هزار و صدگان با صدگان هزار، جابه جا می‌شوند مجموع دو رقم صدگان و هزارگان در عدد جدید، چند است؟

(۱). ۹

(۲). ۵

(۳). ۸

(۴). ۴

In the number 195483, the units digit swaps places with the thousands digit, the tens digit swaps places with the tens of thousands digit, and the hundreds digit swaps places with the hundreds of thousands digit. What is the sum of the two new digits in the hundreds and thousands places in the new number?

1. 9
2. 8
3. 5
4. 4

Mathematics — LPS — Trap:- — Difficult

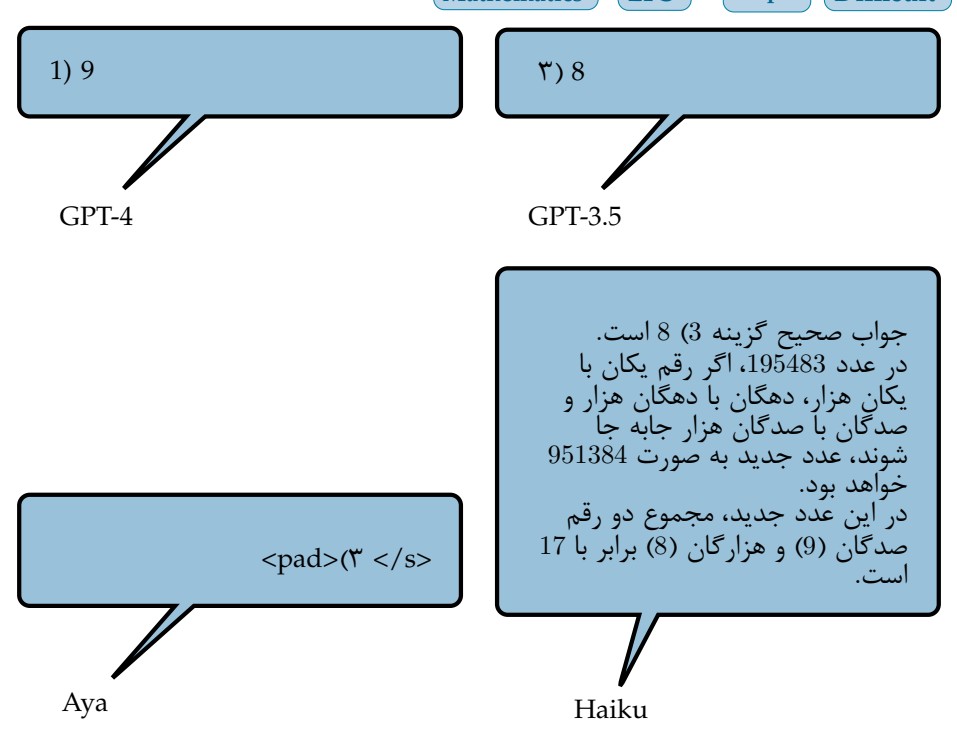

چند مورد از مطالب زیر درست است؟ (عدد اتمی آهن برابر ۶۲ است.)

(آ) کاتیون سازنده نمک به‌دست آمده از واکنش زنگ آهن با هیدروکلریک‌اسید، در آرایش الکترونی خود یک زیرلایه نیمه‌پر دارد.

(ب) تفاوت شمار پیوندهای کووالانسی در ششمین عضو خانواده آلکین‌ها و ششمین عضو خانواده آلکان‌ها برابر با ۲ است.

(پ) سرعت متوسط تولید $HF$ در واکنش $\ll SF_\text{۴} + \text{۲}H_\text{۲}O \rightarrow SO_\text{۲} + \text{۴}HF \gg$ برحسب $mol.\min^{-\text{۱}}.L^{-\text{۱}}$، ۵۱ برابر سرعت متوسط مصرف $SF_\text{۴}$ برحسب $mol.s^{-\text{۱}}$ است. (حجم ظرف واکنش برابر ۱۶$L$ است.)

(ت) عنصر $_\text{۳۲}Ge$ متعلق به دسته‌ای از مواد است که در صنعت الکترونیک کاربرد فراوانی دارند.

(۱). ۱

(۲). ۲

(۳). ۳

(۴). ۴

How many of the following statements are correct? (The atomic number of iron is 26.)

(a) The constituent cation of the salt obtained from the reaction of iron with hydrochloric acid has a half-filled sublayer in its electron configuration.

(b) The difference in the number of covalent bonds in the sixth member of the alkynes family and the sixth member of the alkanes family is equal to 2.

(c) The average rate of production of $HF$ in the reaction $\ll SF4 + 2H2O \rightarrow SO2 + 4HF \gg$ in terms of $mol.min^{-1}.L^{-1}$ is 15 times the average rate of consumption of $SF4$ in terms of $mol.s^{-1}$. (The volume of the reaction vessel is $16L$.)

(d) Element $_{32}Ge$ belongs to a category of substances that are widely used in the electronics industry.

1. 1
2. 2
3. 3
4. 4

Chemistry — USS — Trap: 1 — Difficult

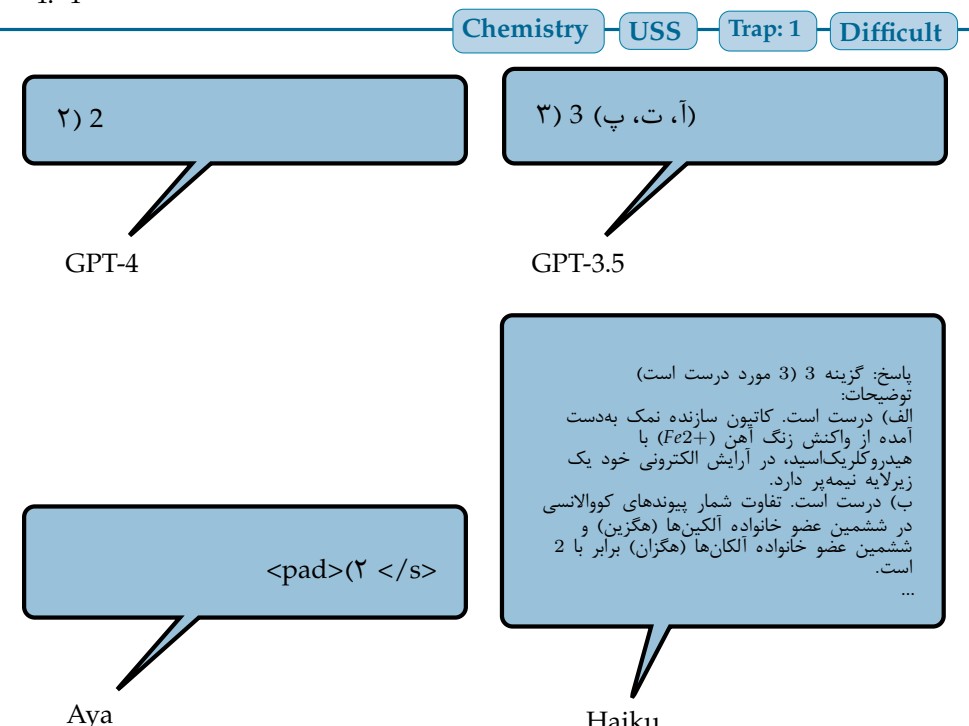

۲) 2

GPT-4

(آ، ت، پ) 3 (۳

GPT-3.5

<pad>(۲ 

Aya

پاسخ: گزینه 3 (3 مورد درست است)
توضیحات:
الف) درست است. کاتیون سازنده نمک به‌دست آمده از واکنش زنگ آهن ($Fe2+$) با هیدروکلریک‌اسید، در آرایش الکترونی خود یک زیرلایه نیمه‌پر دارد.
ب) درست است. تفاوت شمار پیوندهای کووالانسی در ششمین عضو خانواده آلکین‌ها (هگزین) و ششمین عضو خانواده آلکان‌ها (هگزان) برابر با 2 است.
...

Haiku

طبق نظریه اخلاقی افلاطون، انسان نیکبخت کسی است که:

(۱). از میان قوای خود، تنها به قوه عقل که عالی‌ترین قوه است، توجه کند.

(۲). به چهار فضیلت حکمت، خویشتن‌داری، شجاعت و عدالت آراسته گردد.

(۳). از امیال و افعال طبیعی دوری کرده و یکسره به انجام فضیلت‌ها بپردازد.

(۴). عقل او در مقابل قوای شهوت و غضب ایستادگی کرده و اداره بدن را به‌عهده بگیرد.

According to Plato's ethical theory, a fortunate person is one who:

1. Focuses only on the power of reason, which is the highest faculty, among their faculties.

2. Embodies the four virtues of wisdom, self-discipline, courage, and justice.

3. Avoids natural desires and actions and solely engages in virtuous deeds.

4. Their reason stands firm against the forces of lust and anger, taking charge of managing the body.

Philosophy — USS — Trap: - — Difficult

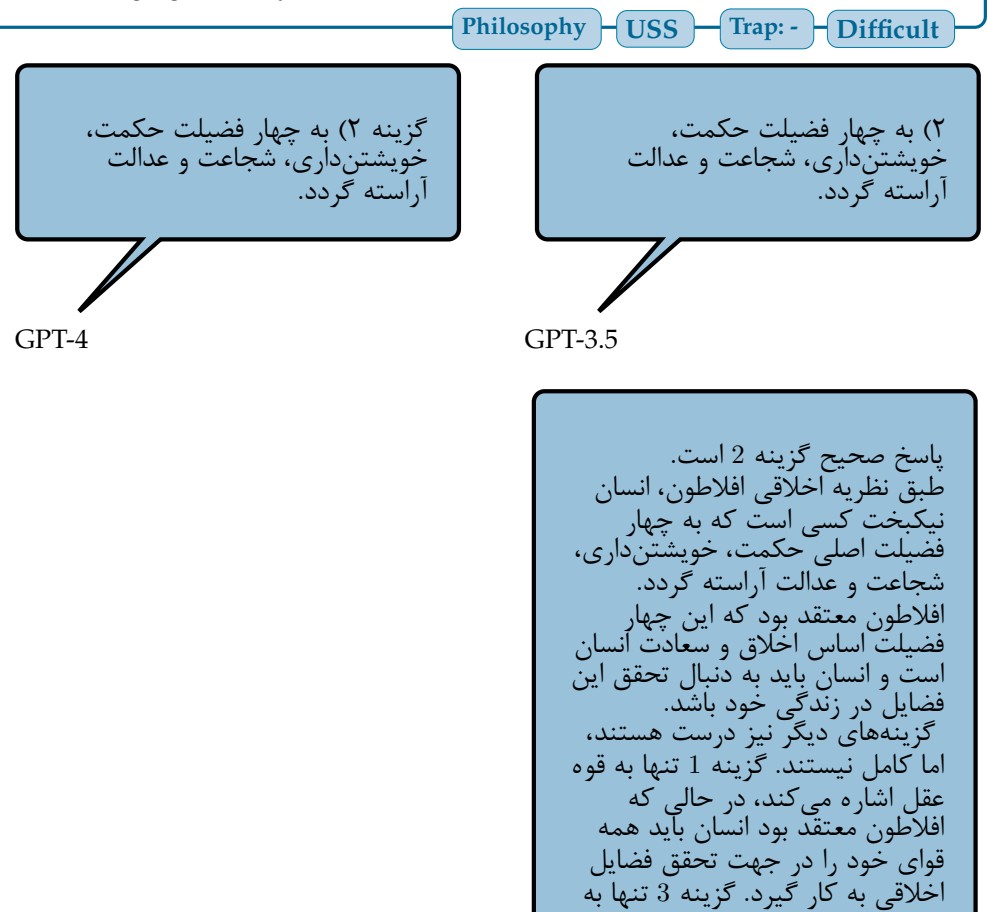

گزینه ۲) به چهار فضیلت حکمت، خویشتن‌داری، شجاعت و عدالت آراسته گردد.

GPT-4

۲) به چهار فضیلت حکمت، خویشتن‌داری، شجاعت و عدالت آراسته گردد.

GPT-3.5

پاسخ صحیح گزینه ۲ است. طبق نظریه اخلاقی افلاطون، انسان نیکبخت کسی است که به چهار فضیلت اصلی حکمت، خویشتن‌داری، شجاعت و عدالت آراسته گردد. افلاطون معتقد بود که این چهار فضیلت اساس اخلاق و سعادت انسان است و انسان باید به دنبال تحقق این فضایل در زندگی خود باشد. گزینه‌های دیگر نیز درست هستند، اما کامل نیستند. گزینه ۱ تنها به قوه عقل اشاره می‌کند، در حالی که افلاطون معتقد بود انسان باید همه قوای خود را در جهت تحقق فضایل اخلاقی به کار گیرد. گزینه ۳ تنها به دوری از امیال و افعال طبیعی اشاره می‌کند، در حالی که افلاطون معتقد بود انسان باید به دنبال تحقق فضایل اخلاقی باشد. گزینه ۴ نیز تنها به نقش عقل در اداره بدن اشاره می‌کند، در حالی که افلاطون معتقد بود انسان باید همه قوای خود را در جهت تحقق فضایل اخلاقی به کار گیرد.

<pad>(۲ 

Aya

46    Haiku

دو کره فلزی خیلی کوچک و مشابه دارای بار الکتریکی ناهمنام $0 > q_1$ و $q_1 > |q_2|$ هستند و در فاصله ۶۰ سانتی‌متری هم قرار دارند و بر هم نیروی الکتریکی $0/9 N$ وارد می‌کنند. اگر کره‌ها را به هم تماس دهیم و دوباره به همان فاصله قبلی از هم دور کنیم، نیروی الکتریکی $1/6$ نیوتون به هم وارد می‌کنند. $q_1$ چند میکروکولن است؟ ($k = 9 \times 10^9 \frac{N.m^2}{C^2}$)

۱ .(۱)

۲ .(۲)

۰/۱ .(۳)

۰/۲ .(۴)

Two very small and similar metallic spheres with unlike electric charges $q_1 > 0$ and $|q_2| > |q_1|$ are placed 60 centimeters apart and exert an electric force of $0.9\,N$ on each other. If we bring the spheres into contact and then separate them again to the same previous distance, they exert a force of $1.6\,N$ on each other. What is the value of $q_1$ in microcoulombs? ($k = 9 \times 10^9 \frac{N \cdot m^2}{C^2}$)

1. 1
2. 2
3. 10
4. 20

Physics — USS — Trap:- — Difficult

---

2 گزینه (۲)

GPT-4

---

۳) 10

GPT-3.5

---

<pad>(۲ 

Aya

---

با توجه به اطلاعات داده شده در سوال، می‌توانیم مقدار بار الکتریکی $q_1$ را محاسبه کنیم.

قانون کولن برای نیروی الکتریکی بین دو بار الکتریکی به صورت زیر است:
$$F = k\frac{q_1 q_2}{r^2}$$

در سوال اول، نیروی الکتریکی بین دو بار الکتریکی برابر با 0.9 نیوتون است و فاصله بین آنها 60 سانتی‌متر است. $0.9 = k\frac{q_1 q_2}{(0.6)^2}$

در سوال دوم، نیروی الکتریکی بین دو بار الکتریکی برابر با 1.6 نیوتون است و فاصله بین آنها همچنان 60 سانتی‌متر است. $1.6 = k\frac{q_1 q_2}{(0.6)^2}$

با حل این دو معادله همزمان، می‌توان مقدار $q_1$ را به دست آورد: $q_1 = 10$ میکروکولن بنابراین، گزینه صحیح گزینه ۳) است.

Haiku

استدلال «هر کلاغی سیاه است، زیرا تا به حال نشنیده و ندیده‌ایم که کلاغی به رنگ دیگر باشد» یک
استدلال . . . است که در آن از . . . نتیجه‌گیری . . . شده است.

(۱). استقرای تعمیمی- موارد جزئی - کلی

(۲). استقرای تعمیمی - موارد کلی - جزئی

(۳). استدلال قیاسی - مقدمات جزئی - کلی

(۴). استدلال قیاسی - مقدمات کلی - جزئی

The argument "Every crow is black because we have never heard or seen a crow of any
other color" is a ... argument, where a ... conclusion is drawn from the ....

1. Inductive generalization - particular instances - general cases

2. Inductive generalization - general instances - particular cases

3. deductive reasoning - specific premises - general cases

4. deductive reasoning - general premises - particular cases

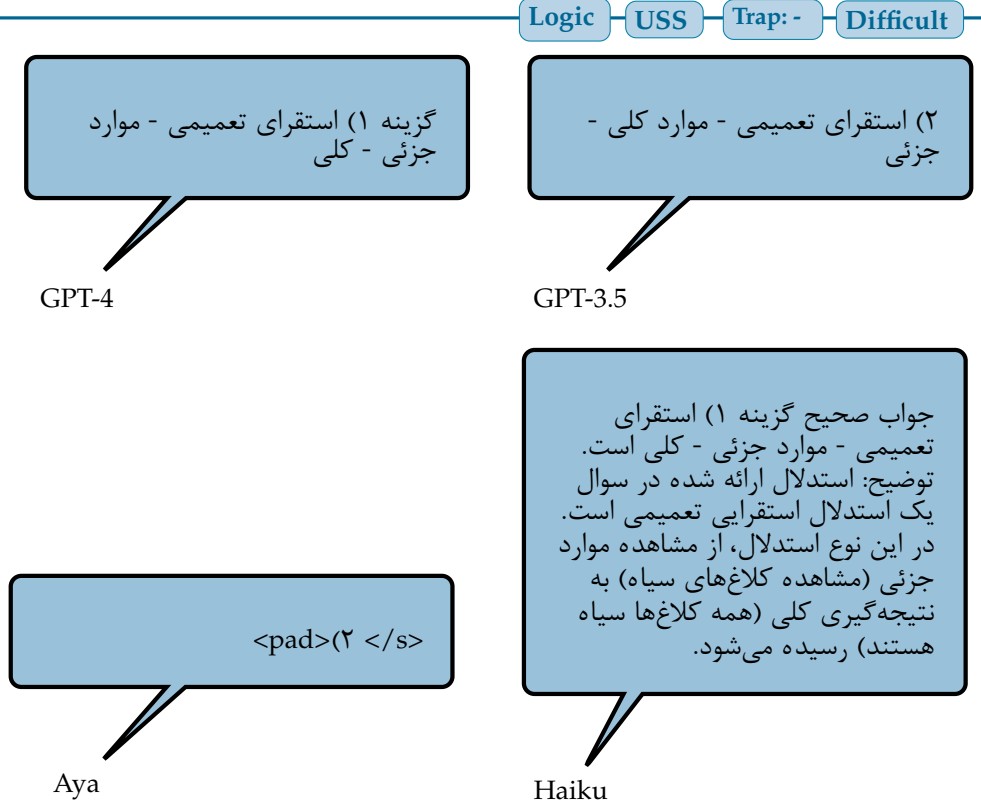

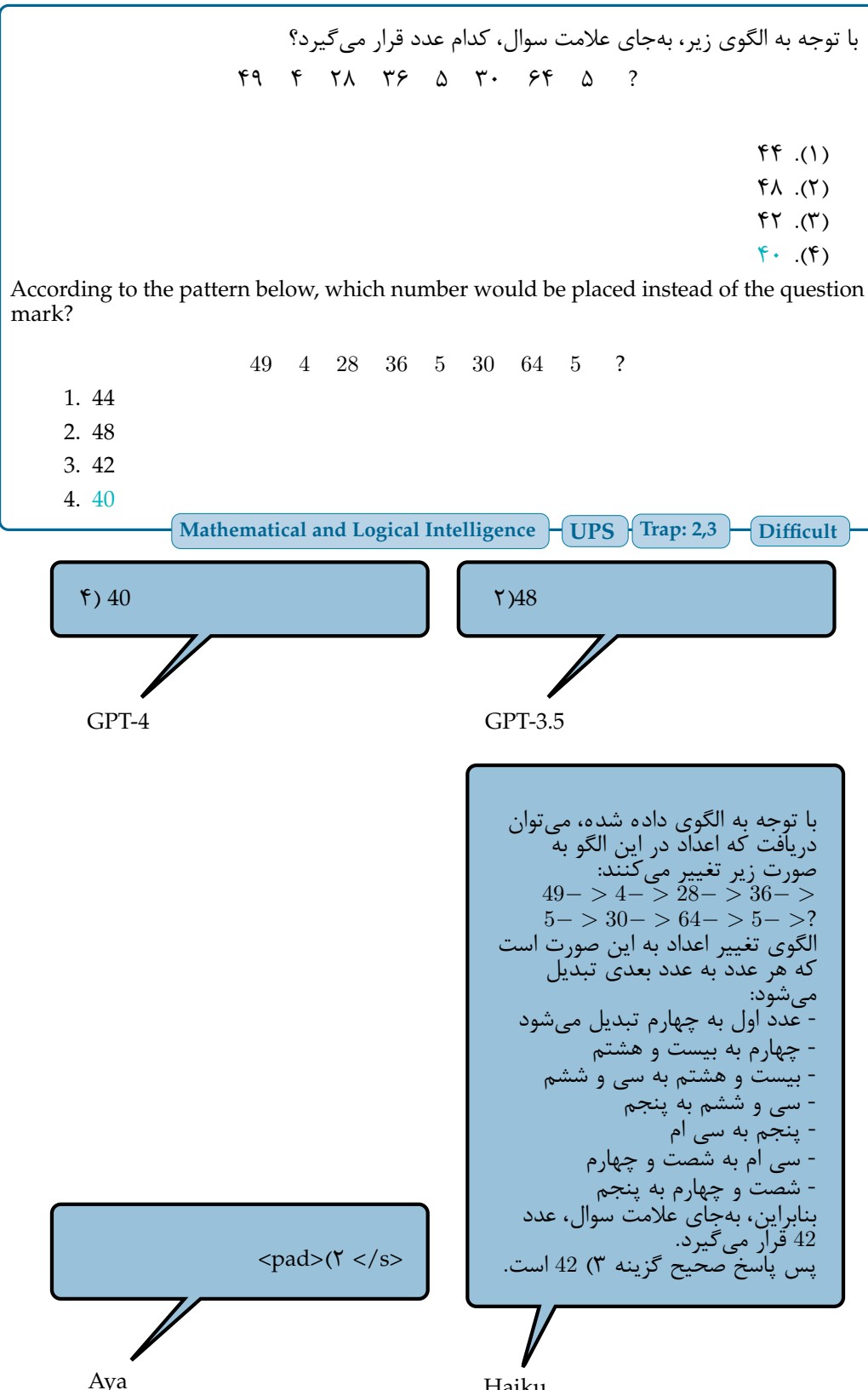

در تمام گزینه‌ها به‌جز ......... رابطه‌ی میان هر دو جفت واژه یکسان است.

(۱). کشتی و اتومبیل / شنا و پیاده‌روی

(۲). دستاورد علمی و تحقیق / گنج و رنج

(۳). جلسه و گفت‌وگو / مدرسه و آموزش

(۴). سه‌شنبه و هفته / سرما و زمستان

In all options except .......... the relationship between both pairs of words is the same.

1. Boats and Cars / Swimming and Walking
2. Scientific achievement and Research / Treasure and Pain
3. Meeting and Discussion / School and Education
4. Tuesday and Week / Cold and Winter

**Verbal and Linguistic Intelligence** — **UPS** — **Trap: 1** — **Difficult**

---

گزینه ۲) دستاورد علمی و تحقیق / گنج و رنج

**GPT-4**

---

۴) سه‌شنبه و هفته / سرما و زمستان

**GPT-3.5**

---

پاسخ: گزینه 1
در گزینه 1، رابطه میان دو جفت واژه متفاوت است. در سایر گزینه‌ها، رابطه میان هر دو جفت واژه یکسان است:
گزینه 2: دستاورد علمی و تحقیق / گنج و رنج (هر دو جفت واژه در رابطه علت و معلولی هستند)
گزینه 3: جلسه و گفت‌وگو / مدرسه و آموزش (هر دو جفت واژه در رابطه مکان و فعالیت هستند)
گزینه 4: سه‌شنبه و هفته / سرما و زمستان (هر دو جفت واژه در رابطه جزء و کل هستند)
اما در گزینه 1، رابطه میان " کشتی و اتومبیل " با " شنا و پیاده‌روی " متفاوت است. در اولی رابطه وسیله و فعالیت است، اما در دومی رابطه فعالیت‌های متفاوت است.

**Haiku**

---

گزینه ۳) جلسه و گفت وگو /pad> مدرسه و آموزش رابطه ی میان هر دو جفت واژه یکسان است.

**Aya**

# I  Overview

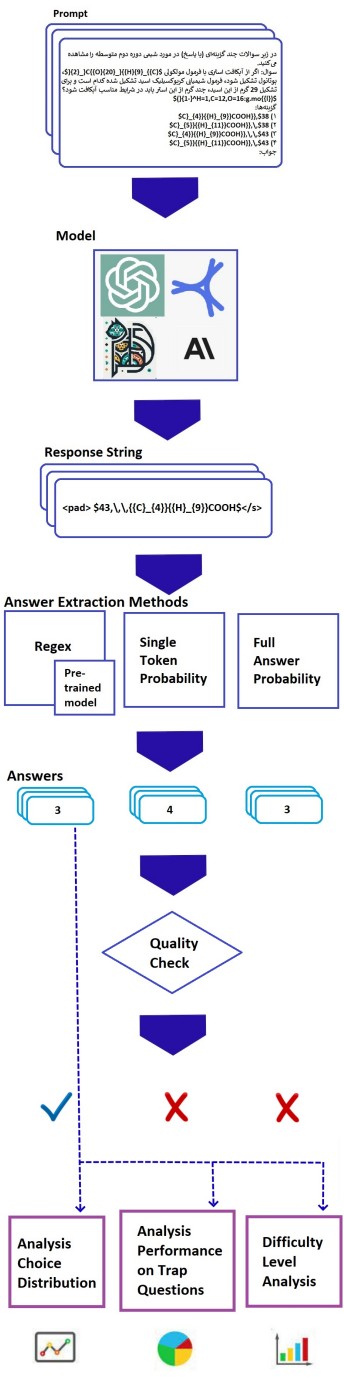

Figure 16: Overview of Khayyam dataset process and analysis

