# OpenReview forum: "Khayyam Challenge (PersianMMLU): Is Your LLM Truly Wise to The Persian Language?"
_colmweb.org/COLM/2024/Conference — COLM_

### Official Review · Reviewer_oJHg · 2024-05-10

**Rating:** 7
**Confidence:** 4
**Ethics Flag:** 1

**Summary:**

This paper describes a new QA dataset for the Persian language. Just like the MMLU dataset for English, this new 'PersianMMLU' dataset covers a wide range of subjects, ranging from the natural sciences to the humanities, and contains more than 20,000 questions. The questions are multiple-choice with 4 answer alternatives, of which one is correct. Each question has its associated metadata, which includes the educational stage for which the question is intended, the difficulty level (which I assume is relative to the intended user group), and a detailed explanation of why the indicated correct answer is correct. The latter is very useful, as it can be used for 'chain-of-thought' training. As a motivation for their work, the authors convincingly point out several shortcomings of using an automatically translated variant of English datasets like the MMLU. The paper also contains experiments of applying various models to answer the questions. An extensive appendix gives lots of details of various aspects of the dataset.

Unless I missed it, it is not explicitly stated in the paper that the dataset will be made public upon acceptance of the paper. It is merely said that 'all associated code is publicly accessible in [anonymous link]'. I assume the dataset will also be published open to everyone, but the authors should state this explicitly.

**Questions To Authors:**

Remove this sales talk, as it is completely unscientific:
"Since conducting its first examination in 1993, Kanoon has been at the forefront of educational innovation, serving a nationwide network of 450,000 students. The center is highly regarded for its comprehensive range of educational services, notably its facilitation of creating and administering smart, customized tests and providing standardized solutions for exercises through the Learning Ladder platform. This
initiative aims to enable educators to design and administer quality, customized tests effortlessly and to provide students with insightful feedback to enhance their learning process."

 "The quality of the questions in the dataset is exceptionally high, a testament to the institution’s reputable standing in the educational sector" How do you measure the quality of the questions? This just seems like more sales talk. Remove.

"The ongoing development of new questions for various educational stages and subjects allows for the continuous expansion and updating of the dataset, reducing the risks of data contamination and erosion." Are you committing to a regular update of the dataset, or what does this mean, really?

How was the human success rate in Figure 2 computed?

Explain the regex method better. Give examples of answers where it is successful. The "single probability probability" and "full answer probability" were also a bit obscure to me. Shouldn't the model just output a single letter ("a", "b", "c", or "d") indicating the correct answer?

"In cases where Regex failed to identify the correct option, we utilized a pre-trained model to generate embeddings for each choice..."  Is this still part of the Regex method, or is this an alternative method?

I didn't understand this sentence, please explain: "We chose the answer selected by humans if the combined total of the remaining three options was lower."

**Reasons To Accept:**

The dataset seems to be a major contribution to Persian NLP. The paper is well written. Everything related to the dataset is clear, but I have some questions about the experiment section.

**Reasons To Reject:**

Unless the authors commit to publishing the dataset free to the world upon acceptance of the paper, the paper should be rejected. I believe the area chair should sort this out with the authors before deciding on publication.

I didn't understand the experimentation section very well. See questions below.

---

> ### Author Rebuttal · Authors · 2024-05-31
>
> We appreciate the reviewer's recognition of this dataset as a major contribution to Persian NLP, as well as the value of its metadata. Below, we provide clarifications and answers to the reviewer's questions in each section.
>
> **Reasons To Reject: Publish the dataset**
>
> For clarification, we commit to fully publish the dataset for free under the terms of the CC BY-ND license.
>
> **Questions To Authors**
>
> 1. *1st and 2nd paragraphs: Remove promotional sentences*
>
> R4.1 We agree that the current wording may be interpreted as promotional. To address this, we have revised Section 3.1 to focus on factual details about Cultural Educational Center's operations relevant to the dataset.
>
> 2. *3rd paragraph: Annual update of the dataset*
>
> R4.2 We plan to update our dataset annually with new samples and insights. This ensures that our dataset remains up-to-date and relevant.
>
> 3. *4th and last paragraphs: Compute human performance*
>
> R4.3 We thank the reviewer. This point has been addressed in R2.1 (reviewer tpuE, answer to item 1).
>
> 4. *5th and 6th paragraphs: Answer extraction methods*
>
> R4.4 The input prompt did not restrict the model to single-letter answers, as it might hinder the reasoning process. Therefore, we used three methods to extract the desired answer from the model output detailed as follows:
>
> The "Regex" method employs regex patterns to extract answer choices (step 1). If this fails, a pre-trained language model generates embedding vectors for each answer choice and the model output. Then, the option with the most similar vector to the output is selected (step 2). Therefore, incorporating a pre-trained language model is part of the "Regex" method. For example, the regex patterns can extract answer choice from "c) 3π" at step 1, but it will fail with "According to the given equation, the third option is correct," requiring step 2.
>
> The "Single Token Probability" method uses the probability distribution for the initial token in the model's output to extract the probabilities associated with answer choice labels (e.g., 'a', 'b', 'c', 'd' in prompt-1 template). The label with the highest probability is selected, leading to the choice of its corresponding option.
>
> The "Full Answer Probability" method utilizes the model's output initial token probability distribution to compute the average log of probabilities for each token of all answer choices. It selects the option with the highest average.
>
> A more detailed explanation is incorporated in the revised version.

---

> > ### Comment · Reviewer_oJHg · 2024-06-05
> >
> > Thanks for the explanations. Great that the dataset will be accessible!

---

### Official Review · Reviewer_yLgN · 2024-05-10

**Rating:** 7
**Confidence:** 5
**Ethics Flag:** 1

**Summary:**

The paper introduces PersianMMLU which consists of a large number of multiple choice questions spanning a wide number of diverse tasks and subjects. The dataset notably does *not* use machine translation (as it is sourced from Persian examinations). The authors surface 3 main contributions: (i) comprehensive coverage of topics (ii) richness of associated metadata (iii) usage of new data to prevent contamination (though to be fair what was new at the time of experimentation is now no longer new in today's vastly connected & scrapable world) (iv) native Persian artifacts (v) inherent scalability.

Overall, the paper is well-written, sets clear differentiation from existing Persian/Farsi datasets like ParSQuAD and ParsiNLU, and presents some metrics on existing LLMs on their new dataset.

**Questions To Authors:**

Section 3.2 talks about the ways the dataset was augmented -- were these all done with human experts or were some extracted from original metadata from learning ladder? For example "Difficulty Level" specifies that each question was classified into one of five distinct difficulty levels but authors don't mention how or by whom.

The dataset seems to be heavily based on data from the Learning Ladder website. Were there additional processing steps that the authors have implemented? Overall, the dataset construction section (3.1) feels a tad sparse.

I did find that mT0XL outperforming a persian-english LLM to be interesting, especially given it's fairly strong performance on other MCQA datasets like Belebele (wondering if the authors dug deeper into this)?

Were the authors able to do any finetuning experiments? E.g. finetuning on MCQA (maybe even Persian MCQA) datasets and then running the model on this particular task?

**Reasons To Accept:**

I appreciate the various experiments that the authors performed including benchmarking various settings like different answer extraction methods, the impact of translation quality, and how few-shot examples did not help on the given MCQA task. The results showing that regex answer extraction outperformed the other methods was not surprising.

**Reasons To Reject:**

I have no reason to reject this paper but I do have a lot of questions.

---

> ### Author Rebuttal · Authors · 2024-05-31
>
> Thank you for the thorough reviews. We are pleased that you appreciated the comprehensive coverage and richness of metadata in PersianMMLU, as well as our experiments benchmarking various settings - key strengths of our work.
>
> Regarding your questions:
>
> 1. *Section 3.2 talks about the ways the dataset was augmented -- were these all done with human experts or were some extracted from original metadata from the learning ladder?*
>
> R3.1 We will clarify in Section 3.2 that all metadata attributes, including difficulty level, were extracted from Learning Ladder. Based on our investigation, Learning Ladder automatically classifies question difficulty using student performance data, marking questions answered incorrectly by more students as difficult. They also determine the "trap" attribute, identifying common wrong answers, by analyzing frequently selected incorrect options.
>
> 2. *The dataset seems to be heavily based on data from the Learning Ladder website. Were there additional processing steps that the authors have implemented.*
>
> R3.2 You're correct that the data is primarily from Learning Ladder. It's worth noting that the Cultural Educational Center ensures the quality of the questions through student assessment, and a team of question designers and editors. However, we further refined the raw data by:
> - Parsing HTML to extract relevant attributes
> - Filtering out questions containing images or tables
> - Removing explanatory answer questions to maintain multiple-choice format consistency
> - Deduplicating questions
>
> We will expand Section 3.1 with more details on these steps.
>
> 3. *I did find that mT0XL outperforming a persian-english LLM to be interesting, especially given it's fairly strong performance on other MCQA datasets like Belebele (wondering if the authors dug deeper into this)?*
>
> R3.3 We agree this is an intriguing result worthy of further investigation. Although examining the specific factors was beyond our current scope, it's possible that the amount of Persian pretraining data for PersianMind (2 billion tokens) may not have been sufficient.
>
> 4. *Were the authors able to do any fine tuning experiments?*
>
> R3.4 We did not fine-tune in this work, as our primary goal was to evaluate models' inherent zero-shot reasoning capabilities. However, fine-tuning, especially on other Persian MCQA datasets, is a logical next step. We will note this as future work.
>
> Thank you again for your constructive feedback.

---

> > ### Comment · Reviewer_yLgN · 2024-06-05
> >
> > Thank you for your rebuttal. I acknowledge having read it.

---

### Official Review · Reviewer_tpuE · 2024-05-11

**Rating:** 6
**Confidence:** 4
**Ethics Flag:** 1

**Summary:**

This work introduces a benchmark akin to MMLU for evaluating and tracking the progress of LLMs for the Persian language. The benchmark is built by gathering the questions from the website of a private educational institution in Iran, which most notably designs and holds regular mock examination sessions for students at different education levels. The benchmark covers 38 subjects and comes with metadata (details in reasons to accept) that facilitates further analysis of the patterns in capabilities of LLMs.

**Questions To Authors:**

Following the previous part of the review, I also have the questions below:
- What does "We chose the answer selected by humans if the combined total of the remaining three options was lower." in 4.2 mean? Does this mean that in the regex method in some cases it was just assumed that the model would go with the same answer as the human?
- Why do the prompts provided in the appendix don't use the same terminology for the grade levels as the main text? For instance, using second "grade of high school" instead of "upper secondary school". Or is that an effect of using machine translation systems?
- I'm not able to see the results corresponding to "4.3 Impact of translation quality" anywhere. Assessing the claims in that section without seeing qualitative results is not viable.

**Reasons To Accept:**

I agree with the authors' argument on the importance of creating language-specific benchmarks rather than translating an English benchmark. So I consider this a huge strength. Additionally, I really like the metadata that's naturally included. The private educational institute that the questions are gathered from provides solution booklets after each session of the regularly held mock exams. The solution booklet includes metadata such as the grade for which the question was designed, its difficulty, descriptive solution, whether or not the question included a trap choice among the choices, students' correct answer rate (correlated with human performance), and the year that the question was designed. I think this metadata is very valuable and can lead to a discussion community on the benefits of including such metadata in creating benchmarks.

**Reasons To Reject:**

Given that this data was collected from a private educational institute, I believe more details should be provided on permission to collect and licensing details. Additionally, I'm not fully sure I understand the answer extraction methods in 4.2. More clear details would be beneficial (also refer to the questions below).

---

> ### Author Rebuttal · Authors · 2024-05-31
>
> We appreciate the reviewer's recognition of the importance of language-specific benchmarks and their view on the value of metadata, including descriptive answers that foster community discussion.
>
> We understand the concern about dataset privacy and publication. This issue has been already discussed with the data owner organization and necessary permission has been obtained for non-commercial use. The PersianMMLU dataset is distributed under the CC BY-ND license.
>
> Regarding the reviewer's questions:
>
> 1. *What does 'We chose the answer selected by humans if the combined total of the remaining three options was lower' in 4.2 means? Does this mean that in the regex method, in some cases it was just assumed that the model would go with the same answer as the human?*
>
>   R2.1 In our dataset, we provide the percentage of respondents who chose each answer option for every question. We label a question as correctly answered by humans if the percentage selecting the correct answer is greater than the combined percentages of the other options; otherwise, we label it as incorrectly answered. Human accuracy is assessed by counting the number of correctly and incorrectly answered questions.
>
> 2. *Why do the prompts provided in the appendix not use the same terminology for the grade levels as the main text?*
>
> R2.2 We thank the reviewer for their precise review. As noted in the figure captions, the English machine translation was provided only for readability purposes. In response to this comment, we have made the translations consistent and corrected them in the revised paper.
>
> 3. *I'm not able to see the results corresponding to '4.3 Impact of translation quality' anywhere. Assessing the claims in that section without seeing qualitative results is not viable.*
>
>   R2.3 Thanks for pointing this out. We will include the relevant results in the camera-ready version. Our results show that model performance on automatic translations improves when high-quality translations (denoted with an asterisk in the following table) are used instead of automatic translations. Specifically, model performance on automatic translations improves for samples in three major subjects—Physics, Math, and Chemistry—by 18%, 2%, and 12% respectively when we use high-quality translations. The translation results are as follows:
>
>        En2Fa | Fa2En
>        En  Fa| En  En* Fa
>        Physics| 42  26|Physics| 58  76  50
>        Biology| 76  74|Chem| 74  82  86
>        ML| 46  40|Math| 50  52  56

---

> > ### Comment · Reviewer_tpuE · 2024-06-07
> > **Response to the Rebuttal**
> >
> > I acknowledge reading the authors' response and thank them for addressing my questions. l believe my original scorer still correctly reflects my overall evaluation of the paper, and therefore, I'm keeping my score the same. I still believe that the "answer extraction methods" need to be described more clearly and recommend that as a major area of focus and perhaps rewriting for the camera-ready version to the authors.

---

### Official Review · Reviewer_WmnJ · 2024-05-13

**Rating:** 6
**Confidence:** 4
**Ethics Flag:** 1

**Summary:**

This paper introduces the Khayyam Challenge (PersianMMLU), a benchmark to evaluate LLMs for the Persian language. It comprises 20k+ multiple choice questions from 38 tasks / subjects, meta information such as difficulty levels and descriptive answers are also included. Several state-of-the-art models including both open-source and commercial models are tested against this newly introduced dataset. Detailed analysis is also conducted to provide new insights.

**Reasons To Accept:**

- Evaluating language models in languages other than English is an important task, and this paper introduces a new benchmark dataset for a relatively less-studied language namely Persian, which would be valuable for future research.
- The newly introduced dataset contains rich metadata, enriching the contribution.
- Multiple multilingual models are employed in the experiments and analysis among different aspects such as trap analysis, selected choice distribution are considered.

**Reasons To Reject:**

- Regarding the data collection process: Since the questions are sourced from a specific website, several potential issues arise, including bias in the questions, contamination of data from a single source, and potential licensing issues. The authors should address these concerns in their clarification.
- The current discussion on data contamination is not convincing; the mere incorporation of previously unused questions does not guarantee the absence of contamination in the dataset. I would suggest employing data contamination detection methods for a more robust analysis.
- While the authors mention the challenge of translating English questions into Persian as a motivation in the introduction, it might be better to explore this issue in the experiments as well. For example, assessing whether translated MMLU can effectively serve as a proxy for testing the model's proficiency in Persian, and identify any associated issues or weaknesses with such translated data.
- A minor point: including a complete example would enhance clarity and provide a more vivid understanding.

---

> ### Author Rebuttal · Authors · 2024-05-31
>
> Thank you for the thorough reviews. We are grateful for highlighting the importance of having evaluation benchmarks for languages other than English, including Persian. The benchmark we introduced offers rich metadata and aims to address this need. We are also grateful that you found our current multilingual assessment and analysis useful.
> Regarding your questions:
>
> 1. *1st paragraph: bias and potential licensing issues*
>
> R1.1 To address concerns of bias, we note that the website serves as a platform for questions contributed by a diverse group of educators from various backgrounds and disciplines in education. This diversity helps to mitigate individual biases. Furthermore, the inclusion of questions from the nationally standardized Konkur exam, which are carefully selected from submissions by top educators, adds an additional layer of quality control. These properties minimize the presence of potential biases in PersianMMLU.
>
> Moreover, the website features an authentication-required login interface that prevents access through common web crawling methods, thereby helping to prevent data contamination and ensuring the integrity of our dataset. Also, regarding the license, the PersianMMLU dataset will be distributed under the terms of the CC BY-ND license.
>
> 2. *2nd paragraph: contamination detection method*
>
> R1.2 We tested the hypothesis that the data from this collection was included in the training data of the models based on paper [1]. We conducted this evaluation using the Aya101 model, the best publicly available model, which had access to the token probabilities. As most contamination detection methods require access to the model's probabilities or weights, which are not available for private/closed models, we used the best performing public model for this evaluation. The hypothesis that the evaluation data was included in the training data was not proven, with a p-value of 0.3909. A detailed explanation of this experiment is included in the supplement of the revised paper.
>
> 3. *3rd paragraph: translation effect analysis*
>
> R1.3 Thank you for pointing this out. This issue has been addressed in our response to Reviewer 2, specifically in section R2.3.
>
> 4. *4th paragraph: example*
>
> R1.4 We have provided a full example in the following anonymous link
>
> https://drive.google.com/file/d/1lgdgb09s4KNbGbfvZk486E6nuGSXEo2T/view?usp=sharing.
>
> [1] https://openreview.net/forum?id=KS8mIvetg2

---

> > ### Comment · Reviewer_WmnJ · 2024-06-05
> >
> > Thanks for your response.
> >
> > Regarding the first point about the "bias" of the dataset: While standard examinations can alleviate some bias, sourcing data from a single source can still introduce significant bias, such as in the style and difficulty of the questions. This is something I would like the authors to discuss in the paper.
> >
> > On the data contamination issue, it's good to see that the authors conducted some numerical checks. However, the hypothesis that "the website features an authentication-required login interface that prevents access through common web crawling methods, thereby helping to prevent data contamination" is not very convincing. I assume there are also many copies outside this specific site if such an exam is a very important one. Therefore, one possible solution would be to check the n-gram overlap of the questions with common crawl dumps, which might be more convincing.
> >
> > Overall, the authors have addressed some of my concerns, and I would like to increase my score to 6.

---

### Decision · Program_Chairs · 2024-07-10

**Decision:**

Accept

**Comment:**

The PersianMMLU dataset is a significant contribution to Persian NLP, mirroring the MMLU dataset for English by covering a wide range of subjects with over 20,000 high-quality multiple-choice questions. The dataset is enriched with valuable metadata, such as educational stage, difficulty level, and detailed explanations, which can aid in 'chain-of-thought' training and further analysis of LLM capabilities. This benchmark dataset addresses the inadequacy of merely translating English datasets and is poised to facilitate meaningful evaluations and advancements in Persian language models. The thorough documentation and inclusion of metadata from a reputable educational institution enhance its utility for the research community.

The paper has several issues that need addressing, primarily concerning the dataset's publication commitment and clarity in the experimental methodology. There is a lack of explicit confirmation that the dataset will be made publicly available, which is crucial for its acceptance. Additionally, the data collection process raises concerns about potential biases, licensing permissions, and data contamination, which the authors have not convincingly addressed. The explanation of the answer extraction methods is unclear and needs further elaboration. The paper also includes unscientific promotional content that detracts from its scholarly value and should be removed. Clarifications on certain experimental procedures and metrics used are necessary for a comprehensive understanding.

[comments from the PCs] Please revise your paper following the critical comments by the author, including improving the scholarly value of your work by removing promotional content, addressing licensing/bias/contamination concerns, and clarifying issues that relate to the data construction method.